# xVerify: Efficient Answer Verifier for Reasoning Model Evaluations

## Abstract

With the release of the o1 model by OpenAI, reasoning models adopting slow thinking strategies have gradually emerged. As the responses generated by such models often include complex reasoning, intermediate steps, and self-reflection, existing evaluation methods and reward models are often inadequate. They struggle to determine whether the LLM output is truly equivalent to the reference answer, and also have difficulty identifying and extracting the final answer from long, complex responses. To address this issue, we propose xVerify, an efficient answer verifier for reasoning model evaluations. xVerify demonstrates strong capability in equivalence judgment, enabling it to effectively determine whether the answers produced by reasoning models are equivalent to reference answers across various types of objective questions. To train and evaluate xVerify, we construct the VAR dataset by collecting question-answer pairs generated by multiple LLMs across various datasets, leveraging multiple reasoning models and challenging evaluation sets designed specifically for reasoning model assessment. A multi-round annotation process is employed to ensure label accuracy. Based on the VAR dataset, we train multiple xVerify models of different scales. In evaluation experiments conducted on both the test set and generalization set, all xVerify models achieve overall F1 scores and accuracy exceeding 95%. Notably, the smallest variant, xVerify-0.5B-I, outperforms all evaluation methods except GPT-4o, while xVerify-3B-Ib surpasses GPT-4o in overall performance. Furthermore, we conduct RL experiments with xVerify as the reward model. Compared with direct generation, it shows an improvement of 18.4% for Qwen2.5-7B. which is greater than when Math Verify is used as the reward function. These results validate the effectiveness and generalizability of xVerify. All resources for xVerify are available at https://anonymous.4open.science/r/xVerify-5702.

## 1 Introduction

With the emergence of chain of thought (CoT) prompting (Wei et al., 2022a), researchers began to explicitly encourage LLMs to generate intermediate reasoning steps, thereby enhancing their ability to handle complex tasks. Following this, OpenAI introduced the o1 model (Jaech et al., 2024), which proposed the concepts of slow thinking and scaling at test time. Specifically, the model is trained to output a detailed reasoning process before generating a final answer, significantly improving its performance on complex tasks. Inspired by this paradigm, a variety of reasoning models have emerged, such as DeepSeek-R1 (DeepSeek-AI et al., 2025) trained with GRPO, OpenAI's o3-mini (OpenAI, 2025), and QwQ-32B (Team, 2025). However, the rise of reasoning models has posed substantial challenges for evaluation. Because the reasoning processes output by these models may include redundant information, intermediate results, and self-reflections, current evaluation methods, as well as reward models in reinforcement learning, often prove ineffective (Chang et al., 2024).

Developing scalable and accurate evaluation methods for LLMs on complex reasoning tasks (e.g., commonsense, logical, multi-hop, and mathematical reasoning) has become increasingly important (Guo et al., 2023). While human annotation remains the gold standard, it is labor-intensive and difficult to scale. Automatic methods fall into two main categories: rule-based frameworks (Open-MMLab, 2023; He et al., 2024; Hynek Kydlíček, 2024; OpenAI, 2024), which extract answers through strict formatting and pattern matching, and LLM-based judge models (Li et al., 2024a; Gu et al., 2025; Li et al., 2025a), which provide qualitative assessments or scores (Chang et al., 2024).

Both types of methods are commonly used to evaluate the performance of final models or to serve as reward models in reinforcement learning. However, rule-based methods struggle with diverse output formats and lengthy chains of thought; for example, Math-Verify (Hynek Kydlíček, 2024), adopted in the Open-R1[1] project, can only handle mathematical results that are strictly formatted and in fixed positions. Although judge models offer adaptability, they are not explicitly trained for objective 'correct/incorrect' decision-making (Gu et al., 2025). Consequently, a robust, automated solution specifically tailored for objective reasoning evaluation is still lacking.

To address these challenges, we introduce xVerify, an efficient LLM-answer verifier tailored for evaluating LLM responses to objective questions. xVerify processes the full LLM output, enabling it to accurately identify final answers from complex reasoning traces. It also supports robust equivalence checking, including symbol conversion (e.g., alpha $\to \alpha$), mathematical expression matching, and semantic alignment in natural language. Moreover, it is tolerant of formatting errors such as malformed LaTeX, making it applicable to a wide range of tasks, including math problems, multiple-choice, short-answer, and classification questions. To train and evaluate xVerify, we construct the **V**erify **A**nswer for **R**easoning (VAR) dataset, which includes responses from 19 LLMs across 24 reasoning benchmarks. All labels are verified through multi-round GPT-4o and human review. The dataset covers advanced reasoning models and benchmarks like GPQA, LiveMathBench, and AIME 2024. We fine-tune xVerify on a variety of base models (e.g., Qwen2.5, LLaMA, Gemma 2) and scales (0.5B–32B). Remarkably, even the smallest variant (xVerify-0.5B-I) surpasses existing evaluation methods—including 32B-sized models—on all metrics, while larger variants achieve F1 and accuracy over 95% on both test and generalization sets. Furthermore, we conduct reinforcement learning (RL) experiments with xVerify as the reward model. Compared with direct generation, it shows an improvement of 18.4% for Qwen2.5-7B. This also represented a greater improvement than when Math Verify is used as the reward model. For Llama3.1-8B, we achieve similar improvements.

The main contributions of this paper are summarized as follows:

- We construct the VAR dataset, which contains answer samples from 19 LLMs across 24 evaluation benchmarks. The dataset is annotated via multiple rounds of GPT-4o and human review, and is designed for training and evaluating judge models for reasoning tasks.

- We propose xVerify, an efficient answer verifier for evaluating reasoning models, and have released several fine-tuned versions that are publicly available on Hugging Face.

- We comprehensively evaluate xVerify in two key capacities: as a judge model, demonstrating superior accuracy and robustness against existing methods on both in-domain and out-of-distribution benchmarks; and as a reward model in RL, where it effectively enhances policy performance compared to direct generation.

## 2 RELATED WORK

Evaluation methods have always been a crucial component in the development of LLM (Chang et al., 2024). However, the open-ended nature of LLM outputs makes it difficult to apply standardized metrics, limiting the effectiveness of traditional evaluation methods (Li et al., 2024a). The rise of reasoning models (OpenAI, 2025; DeepSeek-AI et al., 2025; Team, 2025), which often generate lengthy and complex reasoning, further complicates evaluation. For objective tasks, the main challenge is to accurately extract the final answer from the LLM's semi-structured output and compare it with the reference answer. Existing approaches are typically divided into human evaluation and automatic evaluation. While human evaluation offers flexibility, automatic methods are more cost-efficient and consistent (Chang et al., 2024). Current automatic methods mainly include rule-based evaluation frameworks and LLM-based judgment methods.

Rule-based methods are widely used in automatic evaluation frameworks such as LM Eval Harness (Gao et al., 2021), OpenCompass (OpenMMLab, 2023), UltraEval (He et al., 2024), and OpenAI Evals (OpenAI, 2024). Tools like Math-Verify (Hynek Kydlíček, 2024) also follow this approach, extracting final answers using regular expressions (RegEx) and comparing them with reference answers. However, LLM outputs often contain final answers in varied surface forms—e.g., "alpha" vs. "$\alpha$", "A" vs. "a", or "1000" vs. "$10^3$"—which can be semantically equivalent but textually different. While some tools support limited transformations, they typically handle only LaTeX

---

[1] https://github.com/huggingface/open-r1

expressions or simple string patterns, and struggle with basic semantic equivalence like "one hundred" vs. "100". For reasoning models, the output is usually lengthy and involves complex reasoning steps with intermediate results. This makes it difficult for regular expressions to accurately identify the final answer, causing rule-based approaches to frequently fail in such contexts. Moreover, prior work has shown that LLMs may revise or overturn their initial predictions during extended reasoning processes, exhibiting a kind of self-reflection (Wang et al., 2024a). Additionally, rule-based methods typically ignore the reasoning process and only evaluate the final answer, which has drawn criticism from many researchers—especially in the context of reasoning models (Wei et al., 2022b; Wang et al., 2023; Jacovi et al., 2024). Thus, rule-based evaluations are limited in reasoning scenarios.

LLM-based judgment methods use fine-tuned LLMs to evaluate the quality of other LLMs' responses. Compared to traditional evaluation methods, they offer greater task adaptability, generate interpretable results, reduce evaluation costs, and can be applied across the LLM lifecycle (Li et al., 2024a; Gu et al., 2025; Li et al., 2025a). For objective questions, these judge models can extract final answers from responses with intermediate reasoning or self-reflection. In recent years, many LLM-based judge models have emerged, including JudgeLM (Zhu et al., 2025), PandaLM (Wang et al., 2024b), Auto-J (Li et al., 2024b), Prometheus 2 (Kim et al., 2024), CompassJudger (Cao et al., 2024), CritiqueLLM (Ke et al., 2024), and Themis (Hu et al., 2024). Judge models typically support pointwise, pairwise, and listwise evaluations (Li et al., 2024a), and some also serve as reward models in reinforcement learning. However, most are designed to assign scores to LLM outputs, making them more suitable for subjective evaluations like helpfulness, reliability, or relevance. For objective questions that require binary decisions ("correct" or "incorrect"), these models are less effective. Although scores can be binarized using thresholds, this approach is unreliable, as the models are not explicitly trained for such tasks. Moreover, the current LLM-based critic models and PRMs (Process Reward Models) exhibit subpar performance when detecting errors in long chain-of-thought responses generated by reasoning models (He et al., 2025). Thus, while judge model holds promise for evaluating reasoning models, they require targeted training.

In summary, automatic evaluation on objective tasks remains underdeveloped. Rule-based and LLM-based methods each have clear limitations, while human annotation is costly and hard to scale. To address these challenges, we propose xVerify, a robust and targeted judge model specifically designed for objective evaluation of LLMs.

## 3 PROBLEM DEFINITION

To evaluate the correctness of LLM responses to objective questions, the key is to extract the final answer from the response and compare it with the reference answer. We formally define this evaluation task as follows:

We formalize this task as a 4-tuple $(Q, R, A_{\text{ref}}, E)$, where $Q = \{q_1, q_2, ..., q_n\}$ is the set of questions, $R = \{r_1, r_2, ..., r_n \mid r_i = \mathcal{W}(q_i)\}$ is the set of responses generated by an LLM $\mathcal{W}$, $A_{\text{ref}} = \{a_{ref}^1, ..., a_{ref}^n\}$ is the set of reference answers, and $E : Q \times R \times A_{\text{ref}} \to 0, 1$ is the evaluation function that returns 1 if the response is correct and 0 otherwise.

For the stage of extracting the final answer, given a response $r$ to question $q$, which may include intermediate reasoning and multiple candidate answers, we denote the extracted candidates as $A(r)$. To identify the final answer, we define a scoring function $S : A(r) \times Q \to \mathbb{R}$ that measures the relevance or suitability of each candidate $a \in A(r)$ to $q$, and select the final answer using the extraction function: $\varepsilon(q, r) = \arg\max_{a \in A(r)} S(a, q)$.

For the equivalence comparison stage, we define an equivalence function $\psi : A_{\text{ref}} \times A_{\text{final}} \to \{0, 1\}$, where $\psi$ returns 1 if the predicted answer is equivalent to the reference, and 0 otherwise. Since answers may appear in different forms, $\psi$ integrates results from the following three sub-functions:

For mathematical expressions, we define a composite normalization function $\Phi_{\text{norm}}^{math} = \phi_{\text{err}} \circ \phi_{\text{syn}} \circ \phi_{\text{alg}} \circ \phi_{\text{dim}}$, where $\phi_{\text{err}}$ repairs minor syntax errors, $\phi_{\text{syn}}$ unifies syntactic structures, $\phi_{\text{alg}}$ performs algebraic simplification, and $\phi_{\text{dim}}$ ensures consistency in physical units. By transforming expressions into a canonical form, $\Phi_{\text{norm}}^{math}$ enables reliable equivalence comparison:

$$\psi_{math}(a_{ref}^{math}, a_{final}^{math}) = \begin{cases} 1 & \text{if } \Phi_{\text{norm}}^{math}(a_{ref}^{math}) = \Phi_{\text{norm}}^{math}(a_{final}^{math}), \\ 0 & \text{otherwise} \end{cases} \tag{1}$$

For natural language answers, we define a comparison function $\psi_{\text{nl}} : A_{ref}^{\text{nl}} \times A_{final}^{\text{nl}} \rightarrow \{0, 1\}$ to assess semantic equivalence. Specifically, we introduce a semantic alignment function $\phi_{\text{align}}^{nl}$ to measure the similarity between two textual answers. The equivalence decision is made by comparing the alignment score with a predefined threshold $\tau$:

$$\psi_{nl}(a_{ref}^{nl}, a_{final}^{nl}) = \begin{cases} 1 & \text{if } \phi_{\text{align}}^{nl}(a_{ref}^{nl}, a_{final}^{nl}) \geq \tau, \\ 0 & \text{otherwise} \end{cases} \tag{2}$$

For symbolic representations, we define a composite normalization function $\Phi_{\text{norm}}^{sym} = \phi_{\text{uni}} \circ \phi_{\text{font}} \circ \phi_{\text{dom}}$, which unifies symbols by applying $\phi_{\text{uni}}$ for Unicode normalization, $\phi_{\text{font}}$ for aligning font styles, and $\phi_{\text{dom}}$ for domain-specific mappings. This produces a standardized form for character-level comparison, and the $\Phi_{\text{norm}}^{sym}$ is defined as:

$$\psi_{sym}(a_{ref}^{sym}, a_{final}^{sym}) = \begin{cases} 1 & \text{if } \Phi_{\text{norm}}^{sym}(a_{ref}^{sym}) = \Phi_{\text{norm}}^{sym}(a_{final}^{sym}), \\ 0 & \text{otherwise} \end{cases} \tag{3}$$

Based on the above components, we define a unified equivalence function $\psi$ to determine whether the final answer $a_{final}$ matches the reference answer $a_{ref}$ across different modalities. Defined as:

$$\psi(a_{final}, a_{ref}) = \begin{cases} 1, & \text{if } \psi_{math}(a_{final}^{math}, a_{ref}^{math}) = 1 \\ & \quad \wedge\ \psi_{nl}(a_{final}^{nl}, a_{ref}^{nl}) = 1 \\ & \quad \wedge\ \psi_{sym}(a_{final}^{sym}, a_{ref}^{sym}) = 1; \\ 0, & \text{otherwise} \end{cases} \tag{4}$$

Here, $a_{final}^{math}$, $a_{final}^{nl}$, and $a_{final}^{sym}$ represent the mathematical, natural language, and symbolic parts of the final answer, respectively, and similarly for $a_{ref}$. This allows for equivalence checking in both unimodal and multimodal settings.

To summarize, the overall evaluation function E is defined as: $E(q, r, a_{ref}) = \psi(\varepsilon(q, r),\ a_{ref})$, where $q$ is the objective question, $r$ is the response generated by the LLM, and $a_{ref}$ is the corresponding reference answer.

## 4 METHODOLOGY

The xVerify training and evaluation pipeline includes three main stages: collecting LLM responses, VAR dataset construction, and xVerify judge pipeline (see Figure 1). We first gather question–response pairs from various LLMs across four types of objective questions, including complex, reasoning-intensive examples. To ensure accurate labels, we employ multiple rounds of annotation and rechecking using both GPT-4o and human annotators. We also apply data augmentation to increase the dataset's diversity and complexity. Finally, we train xVerify models of different sizes on the VAR dataset to evaluate long, multi-step answers—cases that are often difficult for existing evaluation methods. Section 4.1 details the dataset construction, and Section 4.2 describes the training process.

### 4.1 VAR DATASET

xVerify is designed to assess the correctness of reasoning models' responses on objective questions. However, current judge models are mostly trained on tasks such as scoring or reviewing, and reasoning models with lengthy responses have only recently emerged. As a result, there is currently no suitable dataset for training xVerify. To better train and evaluate xVerify, we constructed a dedicated dataset named Verify Answer for Reasoning (VAR). Examples from the VAR dataset are provided in Appendix C.3.

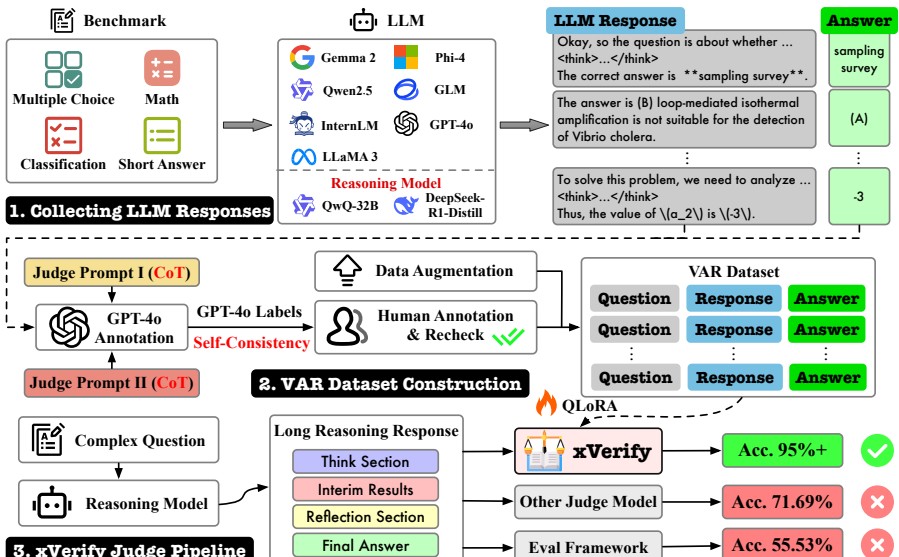

Figure 1: Framework of xVerify: (1) Collecting LLM Responses: aggregate responses from multiple LLMs across datasets covering four question types. (2) VAR Dataset Construction: employ GPT-4o and human annotators for labeling and rechecking, and use data augmentation to refine the dataset. (3) xVerify Judge Pipeline: accurately evaluate multi-component answers from reasoning models on challenging questions.

### 4.1.1 LLM RESPONSE GENERATION

To ensure the diversity and coverage of the dataset, we selected 19 mainstream LLMs and 24 frequently used multilingual datasets to generate and collect responses. To better simulate the answering patterns of reasoning models in common evaluation scenarios, the chosen LLMs include recently released models such as the DeepSeek-R1-Distill series (DeepSeek-AI et al., 2025) and QwQ-32B (Team, 2025). Most of the other LLMs also support context lengths exceeding 32k tokens, enabling them to produce answers with extended reasoning chains. The selected datasets include high-difficulty benchmarks commonly used for evaluating reasoning models, such as GPQA (Rein et al., 2024), AIME 2024 (MAA, 2024), MATH (Hendrycks et al., 2021), and LiveCodeBench (Liu et al., 2024), which typically require multi-step reasoning and computation to solve. During data generation, we also retained some extremely long responses, such as those exceeding 6k characters in length. Detailed information on all LLMs and datasets is provided in Appendix B.

To train and evaluate xVerify more effectively, we grouped the 24 datasets into four types based on question and answer formats: multiple choice, math, short answer, and classification. Multiple choice questions offer several labeled options; math includes questions where answers are mathematical expressions (e.g., numbers, equations), including mathematics and physics problems; short answer questions expect brief natural language responses like names or dates, with no strict format constraints; classification tasks involve selecting the correct label, such as for sentiment or topic classification.

To reflect realistic evaluation settings and generate a diverse set of Q&A samples, we designed multiple prompt templates for guiding the LLMs in response generation. The prompt configurations vary along several dimensions: 0-shot vs. 5-shot, with or without CoT, and with or without answer format restrictions (restrict), resulting in eight distinct prompt types. Details of all prompt templates are provided in Appendix F.1.

In total, we generated 191,600 Q&A samples using the 19 LLMs and 24 evaluation sets, providing a rich and diverse sample pool for constructing the dataset.

### 4.1.2 DATASET PARTITIONING

Based on the previously collected sample pool, we constructed the training, test, and generalization sets through filtering and preprocessing.

The training and test sets are used to train and evaluate the xVerify model. Both are sampled from the same pool, sharing similar distributions. Specifically, they include samples generated by 15 LLMs across 17 evaluation sets, covering the four previously mentioned question types. The training set contains 36,941 samples, and the test set includes 5,194 samples.

The generalization set complements the test set by evaluating xVerify's ability to handle more diverse and challenging distributions, reflecting real-world scenarios. It consists of 5,366 samples from 7 evaluation sets not used in the training or test sets, while still spanning all four question types. These samples are generated by 19 LLMs, including 4 models not seen in training or testing, such as the reasoning model QwQ-32B, resulting in greater diversity and distribution shift.

Section 4.1.4 introduces our data augmentation strategy, which adds more challenging samples to all three sets. Detailed dataset statistics are provided in Appendix C.1.

### 4.1.3 DATA ANNOTATIONS

To ensure the accuracy of xVerify's training and evaluation, we conducted multiple rounds of automatic and manual annotation across the three datasets. Specifically, we used GPT-4o to perform two rounds of annotation for all samples in the datasets, utilizing two distinct prompt templates (details provided in Appendix F.2) to improve annotation confidence (Wang et al., 2023; Liang et al., 2024). Given the large size of the training set, we only applied manual annotation to the more challenging math problems and to samples where the two rounds of GPT-4o annotations disagreed. In contrast, for the test and generalization sets, we manually annotated all samples, resulting in a three-round annotation process to maximize label reliability. Details of the manual annotation process are provided in Appendix C.2.

### 4.1.4 DATA AUGMENTATION

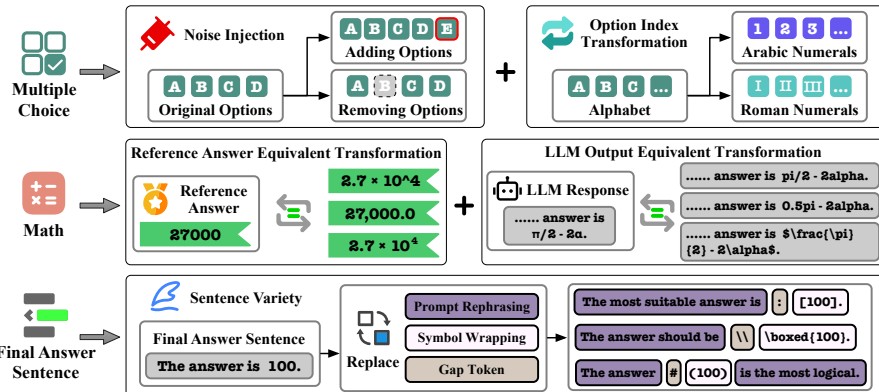

Figure 2: Data Augmentation Pipelines: (1) transformation of multiple-choice options through numbering conversion and noise injection, (2) diversification of mathematical answers via equivalent expression generation, and (3) final answer sentence transformation using prompt rephrasing, symbol wrapping, and gap token insertion.

To further enhance the diversity and robustness of the dataset, we designed a series of data augmentation strategies (illustrated in Figure 2) to better simulate real-world evaluation settings and improve the model's tolerance to varied answer formats.

For multiple-choice questions, we applied two types of augmentations: option index transformation and noise injection. The former converts alphabetical labels to Arabic or Roman numerals, while the latter randomly adds or removes irrelevant distractor options without changing the original question intent, thereby increasing structural complexity.

For math problems, we used two approaches: augmentation based on reference answers and LLM responses. In the first approach, we generated 3–5 mathematically equivalent expressions of each reference answer through symbolic and formal transformations, then created new samples accordingly. In the second, we applied the same transformation logic to the final answers in LLM responses, enriching the dataset with varied mathematical formats and helping the model learn equivalence across symbolic expressions.

We also augmented the final answer statements. Specifically, we extracted answer-bearing sentences from responses generated using restrict prompts, and applied over 1,000 transformation patterns. These included: 20 variations of prompt rephrasing (e.g., "The answer is B" → "The most appropriate answer is B"), 18 symbolic wrappers (e.g., wrapping B as $\boxed{B}$), and 5 forms of delimiter insertions (e.g., adding a colon or space before the answer). This improved diversity in answer formats and reduced overfitting to specific templates.

Together, these strategies expanded the expressive space of the dataset while preserving semantic consistency, offering richer and more challenging training signals for xVerify. After augmentation, the sizes of the training, test, and generalization sets increased to 43,204, 6,122, and 6,468 samples respectively. Full dataset details are provided in Appendix C.1. The augmentation of math problems primarily relied on GPT-4o; prompt templates are listed in Appendix F.3.

## 4.2 MODEL TRAINING

We trained 14 models with different parameter sizes and architectures using the training set from the VAR dataset. Specifically, we utilized the LLaMA-Factory framework (Zheng et al., 2024) and QLoRA technique (Dettmers et al., 2023) for model training. Based on extensive experimentation, we set the number of epochs to 1 and selected a learning rate of 1e-4 as the optimal configuration, with other hyperparameters detailed in Appendix D.1. Many researchers have pointed out potential bias in using LLMs as judge models, where models from the same family tend to receive higher ratings (Li et al., 2025b). To thoroughly evaluate the generalization capability of the xVerify method, we trained 14 models with varying parameter sizes and architectures. These models ranged from 0.5B to 32B parameters and included five different families, such as LLaMA 3 (Grattafiori et al., 2024), Qwen2.5 (Yang et al., 2024), and Gemma 2 (Team et al., 2024). Details of the models used are provided in Appendix D.2.

## 5 EXPERIMENTS

This section details our experiments, covering two main experiments: evaluating xVerify as a judge model on both in-domain and out-of-distribution datasets, and using xVerify as a reward model in the reinforcement learning optimization process. First, we will outline the experimental setup:

- **Datasets:** For the evaluation experiments, we primarily use the test set and generalization set from the VAR dataset. The test set evaluates xVerify's core performance, while the generalization set assesses its robustness on out-of-distribution samples. For the reinforcement learning experiments, the dataset for training and testing the policy model is collected from diverse sources (see Appendix E.2 for details).

- **Metrics:** The main metrics are accuracy, which indicates the overall performance, and F1 score, which combines precision and recall for a more complete perspective.

- **Baselines:** There are two types of baselines: evaluation frameworks and judge models. The evaluation frameworks include DeepSeek-Math (Shao et al., 2024), LM Eval Harness (Gao et al., 2021), Math-Verify (Hynek Kydlíček, 2024), OpenAI Evals (OpenAI, 2024), Open-Compass (OpenMMLab, 2023), and UltraEval (He et al., 2024). The judge models include PandaLM (Wang et al., 2024b), Auto-J (Li et al., 2024b), Prometheus 2 (Kim et al., 2024), JudgeLM (Zhu et al., 2025), and CompassJudger (Cao et al., 2024). In addition, GPT-4o is also used as a judge model with two strategies: one with CoT and one without. The prompts for the judge model and xVerify are provided in Appendix F.4 and Appendix F.5.

### 5.1 EVALUATION WITH XVERIFY AS JUDGE MODEL

Firstly, we evaluated all evaluation frameworks, judge models, and xVerify models on the test and generalization sets of VAR (see Table 1- 2). Overall, the xVerify model demonstrated its effectiveness, consistently achieving the best or second-best values in each column.

**Test Set Evaluation Results.** On the VAR test set, **xVerify consistently outperforms all evaluation frameworks and judge models**. Even the smallest xVerify-0.5B-I achieves second-best overall accuracy (96.85%) and F1 (96.69%), surpassing CompassJudger-1-32B and matching GPT-4o's performance while using far fewer tokens. Larger xVerify variants (3B–32B) further improve

Table 1: Evaluation Accuracy Results on the Test Set. "-" indicates that the evaluation method is not applicable to the problem type. The best performance in each column will be shown in **bold**, and the second-best performance will be underlined.

| Method Type | Method | Multiple Choice | | Math | | Short Answer | | Classification | | Overall | |
|---|---|---|---|---|---|---|---|---|---|---|---|
| | | F1 | Acc. | F1 | Acc. | F1 | Acc. | F1 | Acc. | F1 | Acc. |
| Evaluation Framework | DeepSeek Math Verify | 70.77% | 75.17% | 78.34% | 84.30% | - | - | - | - | 74.90% | 52.52% |
| | LM Eval Harness | 58.44% | 68.19% | 25.16% | 28.27% | 53.41% | 44.51% | 72.35% | 66.94% | 47.67% | 48.32% |
| | Math-Verify | 5.88% | 53.76% | 82.55% | 86.70% | 42.27% | 71.91% | 0.00% | 29.66% | 45.64% | 65.91% |
| | OpenAI Simple Evals | 23.61% | 28.02% | 66.79% | 76.88% | 42.23% | 55.32% | 73.29% | 67.87% | 51.17% | 58.10% |
| | OpenCompass | 68.11% | 72.52% | 79.25% | 84.73% | - | - | - | - | 74.18% | 79.64% |
| | UltraEval | 17.34% | 18.04% | 8.88% | 56.89% | - | - | - | - | 13.95% | 40.71% |
| Judge Model | PandaLM-7B-v1 | 4.26% | 8.12% | 16.78% | 14.46% | 23.47% | 17.72% | 25.32% | 16.79% | 16.40% | 13.72% |
| | Auto-J-13B | 40.00% | 63.20% | 26.32% | 60.62% | 64.41% | 78.22% | 86.04% | 82.60% | 53.38% | 68.13% |
| | Prometheus-8x7B-v2.0 | 71.26% | 68.61% | 71.99% | 66.92% | 76.24% | 77.70% | 83.27% | 77.65% | 74.57% | 71.12% |
| | JudgeLM-13B-v1.0 | 56.81% | 48.89% | 58.39% | 59.46% | 77.32% | 79.52% | 95.63% | 93.82% | 68.57% | 65.83% |
| | JudgeLM-33B-v1.0 | 42.86% | 43.24% | 44.82% | 46.03% | 57.86% | 62.23% | 73.42% | 67.56% | 52.00% | 51.75% |
| | Compass,Judger-1-14B | 58.94% | 44.62% | 55.09% | 40.76% | 59.66% | 52.90% | 90.87% | 86.61% | 63.22% | 51.37% |
| | Compass,Judger-1-32B | 95.09% | 95.37% | 84.11% | 84.30% | 94.95% | 96.11% | 98.45% | 97.84% | 91.67% | 91.69% |
| | GPT-4o as Judge | 96.61% | 96.75% | 95.27% | 95.80% | 95.01% | 96.20% | 98.14% | 97.43% | 96.25% | 96.39% |
| | GPT-4o as Judge (CoT) | 97.10% | 97.23% | 95.41% | 95.88% | 95.63% | 96.63% | 99.56% | 99.38% | 96.85% | 96.95% |
| xVerify | xVerify-0.5B-I | 97.78% | 97.90% | 93.74% | 94.64% | 96.72% | 97.49% | 99.71% | 99.59% | 96.69% | 96.85% |
| | xVerify-3B-Ib | 97.31% | 97.41% | 95.65% | 96.18% | 96.38% | 97.23% | 99.78% | 99.69% | 97.17% | 97.27% |
| | xVerify-7B-I | 97.75% | 97.84% | 95.94% | 96.44% | 96.51% | 97.32% | 99.78% | 99.69% | 97.41% | 97.50% |
| | xVerify-9B-I | 97.43% | 97.53% | 95.75% | 96.27% | 96.06% | 96.97% | 99.78% | 99.69% | 97.19% | 97.29% |
| | xVerify-14B-Ia | 97.49% | 97.59% | 95.73% | 96.22% | 95.41% | 96.46% | 99.63% | 99.49% | 97.06% | 97.16% |
| | xVerify-32B-I | 97.81% | 97.90% | 95.88% | 96.31% | 96.18% | 97.06% | 99.71% | 99.59% | 97.32% | 97.40% |

both F1 and accuracy, peaking at 97.50%/97.41% (F1/Acc.) with xVerify-7B-I. Notably, all xVerify models above 0.5B exceed 95% on challenging math questions, and performance gains taper beyond 7B parameters—suggesting a sweet spot around mid-scale models for this dataset.

Table 2: Evaluation Accuracy Results on the Generalization Set. "-" indicates that the evaluation method is not applicable to the problem type. The best performance in each column will be shown in **bold**, and the second-best performance will be underlined.

| Method Type | Method | Multiple Choice | | Math | | Short Answer | | Classification | | Overall | |
|---|---|---|---|---|---|---|---|---|---|---|---|
| | | F1 | Acc. | F1 | Acc. | F1 | Acc. | F1 | Acc. | F1 | Acc. |
| Evaluation Framework | DeepSeek Math Verify | 72.90% | 73.39% | 11.69% | 79.83% | - | - | - | - | 60.57% | 44.42% |
| | LM Eval Harness | 61.60% | 65.37% | 7.03% | 18.48% | 58.22% | 45.09% | 92.06% | 88.21% | 55.81% | 51.30% |
| | Math-Verify | 5.19% | 45.10% | 64.18% | 87.68% | 9.12% | 52.75% | 0.00% | 24.59% | 16.10% | 55.53% |
| | OpenAI Simple Evals | 28.72% | 29.23% | 24.31% | 78.90% | 58.33% | 59.58% | 94.39% | 91.62% | 57.99% | 63.36% |
| | OpenCompass | 71.64% | 71.44% | 47.22% | 84.39% | - | - | - | - | 65.74% | 78.18% |
| | UltraEval | 16.29% | 15.31% | 13.55% | 78.39% | - | - | - | - | 15.71% | 48.13% |
| Judge Model | PandaLM-7B-v1 | 4.28% | 7.85% | 9.91% | 15.97% | 45.81% | 31.43% | 36.23% | 25.99% | 23.74% | 19.14% |
| | Auto-J-13B | 34.87% | 52.78% | 9.86% | 76.54% | 85.12% | 86.97% | 77.67% | 71.99% | 60.43% | 71.35% |
| | Prometheus-8x7B-v2.0 | 74.13% | 68.60% | 49.48% | 60.27% | 87.15% | 86.13% | 84.70% | 77.19% | 74.51% | 71.69% |
| | JudgeLM-13B-v1.0 | 65.39% | 57.80% | 21.61% | 44.87% | 86.11% | 84.53% | 91.78% | 86.89% | 69.18% | 65.63% |
| | JudgeLM-33B-v1.0 | 46.99% | 45.10% | 20.31% | 39.99% | 71.34% | 66.69% | 41.92% | 33.36% | 46.06% | 46.01% |
| | Compass,Judger-1-14B | 63.65% | 49.50% | 27.63% | 21.20% | 73.61% | 66.48% | 88.97% | 81.92% | 63.10% | 51.21% |
| | Compass,Judger-1-32B | 92.93% | 92.32% | 72.05% | 84.91% | 96.81% | 96.86% | 98.05% | 97.05% | 91.90% | 92.04% |
| | GPT-4o as Judge | 95.86% | 95.38% | 87.91% | 94.76% | 97.46% | 97.49% | 98.67% | 97.98% | 96.03% | 96.18% |
| | GPT-4o as Judge (CoT) | 95.44% | 94.88% | 88.34% | 94.71% | 97.39% | 97.42% | 98.36% | 97.52% | 95.79% | 95.92% |
| xVerify | xVerify-0.5B-I | 96.49% | 96.10% | 80.00% | 91.94% | 96.95% | 97.00% | 99.03% | 98.53% | 95.29% | 95.53% |
| | xVerify-3B-Ib | 96.21% | 95.71% | 86.20% | 94.15% | 97.60% | 97.63% | 99.03% | 98.53% | 96.08% | 96.23% |
| | xVerify-7B-I | 96.16% | 95.66% | 87.86% | 94.87% | 97.45% | 97.49% | 98.93% | 98.37% | 96.22% | 96.37% |
| | xVerify-9B-I | 96.06% | 95.55% | 87.47% | 94.76% | 97.53% | 97.56% | 99.13% | 98.68% | 96.23% | 96.38% |
| | xVerify-14B-Ia | 96.11% | 95.60% | 90.20% | 95.74% | 97.32% | 97.35% | 99.13% | 98.68% | 96.53% | 96.65% |
| | xVerify-32B-I | 96.22% | 95.71% | 90.09% | 95.59% | 97.32% | 97.35% | 99.03% | 98.53% | 96.50% | 96.60% |

**Generalization Set Evaluation Results.** On the VAR generalization set, xVerify's overall F1 and accuracy drop by less than 1.5%, demonstrating strong robustness to out-of-distribution samples. Even xVerify-0.5B-I retains 95.53% accuracy, outperforming all rule-based frameworks and most judge models except GPT-4o. Larger xVerify models reduce the performance gap further: xVerify-14B-Ia reaches 96.65% accuracy with over 90% on math questions. These results confirm that scaling xVerify enhances generalization, and that fine-tuned judge models can outperform CoT-based prompting without incurring extra token costs.

Appendix G provides a comprehensive comparative evaluation of additional models, alongside analyses of their computational efficiency and the evaluation cost of GPT-4o.

In addition to the main experiments, we conducted several supplementary studies to further validate the robustness and design choices of xVerify (see Appendix G.5–G.7). First, a comparison between QLoRA and full fine-tuning showed that QLoRA achieves slightly better performance with significantly lower computational cost, suggesting that full fine-tuning may be more prone to overfitting. Second, systematic evaluation of xVerify against its base models demonstrated substantial

improvements: fine-tuned xVerify variants consistently outperformed their underlying models by more than 10 percentage points in both F1 and accuracy, with many improvements falling in the range of 20–50%. Finally, we verified that xVerify remains highly accurate even when the original question is omitted, indicating that its core judgment mechanism primarily depends on the consistency between the model's answer and the reference answer. Together, these supplementary results highlight the efficiency, effectiveness, and robustness of xVerify across different settings.

## 5.2 REINFORCEMENT LEARNING WITH XVERIFY AS REWARD MODEL

We investigate whether a more accurate reward model improves optimization efficiency and final performance in RL fine-tuning. We build on veRL (Sheng et al., 2024) and train Qwen2.5-7B and Llama3.1-8B with GRPO (Shao et al., 2024), using xVerify-7B-I as the reward and Math-Verify as a rule-based baseline. This experiment is a proof-of-concept designed to compare reward-signal quality and training dynamics rather than to maximize final scores. Training hyperparameters are listed in Appendix E.1. For faithful evaluation, we manually annotate all samples (Appendix C.2).

Table 3 shows that RL with xVerify substantially improves over direct generation (e.g., Qwen2.5-7B gains $18.4\%$ on the seven-benchmark average). Compared to the Math-Verify baseline, xVerify achieves higher final averages, 73.0% versus 72.2% for Qwen2.5-7B and 61.2% versus 60.4% for Llama3.1-8B. The modest improvement is consistent with the constrained proof-of-concept setting and the limited capacity of the policy models. Crucially, the training dynamics highlight xVerify's advantage. The RL learning curves, which plot evaluation accuracy over training steps, show that for both Qwen2.5-7B and Llama3.1-8B, xVerify starts at a higher accuracy and converges in fewer steps than Math-Verify. See the RL learning-curve plots in Appendix G.4. Moreover, Math-Verify induces reward hacking (e.g., consistently appending \boxed{}), whereas xVerify rewards correctness without enforcing brittle formats, leading to more effective learning.

Furthermore, We show that xVerify aligns closely with human judgments, with full results provided in Appendix G.3.

Table 3: Evaluation Accuracy Results of RL with xVerify as Reward Model.

| Model | MMLU-Pro | GPQA | MATH-500 | DROP | Amazon | CMMLU | CHID | Avg. |
|---|---|---|---|---|---|---|---|---|
| **Qwen2.5-7B** | | | | | | | | |
| Direct Generation | 40.4% | 18.8% | 39.6% | 70.4% | 94.0% | 76.8% | 42.4% | 54.6% |
| ↪ RL with Math Verify | 52.0% | **41.2%** | 79.5% | 84.8% | 96.0% | 88.0% | **64.0%** | 72.2% |
| ↪ RL with xVerify | **53.6%** | 39.6% | **81.5%** | **86.0%** | **98.4%** | **88.4%** | 63.6% | **73.0%** |
| **Llama3.1-8B** | | | | | | | | |
| Direct Generation | 42.8% | 28.8% | 52.6% | 82.4% | 96.0% | 55.2% | 42.4% | 57.2% |
| ↪ RL with Math Verify | **46.4%** | 32.4% | 57.9% | **88.8%** | 97.2% | 58.0% | 42.4% | 60.4% |
| ↪ RL with xVerify | **46.4%** | **34.4%** | **58.0%** | 87.6% | **97.6%** | **59.6%** | **44.8%** | **61.2%** |

## 6 CONCLUSION

In this paper, we propose an efficient answer verifier for reasoning model evaluations, named xVerify, which can effectively assess the correctness of long reasoning responses generated by reasoning models on various difficult objective questions. To train and evaluate the xVerify model, we constructed the VAR dataset based on several popular LLMs and evaluation sets. This dataset primarily collects long reasoning responses generated by reasoning models on challenging questions, and multiple rounds of labeling and verification were conducted using GPT-4o and human annotators. Ultimately, we trained multiple xVerify models of varying specifications based on the VAR dataset and performed comparative evaluations with several evaluation frameworks and judge models on both the test and generalization sets. The experimental results show that even the smallest xVerify-0.5B-I model outperforms all methods except GPT-4o, and larger xVerify models surpass all other methods, demonstrating the effectiveness and generalization ability of xVerify. RL experiments demonstrate that xVerify, when is used as a reward model, effectively enhances policy performance compared to direct generation.

## 7 ETHICS STATEMENT

This work complies with the ICLR Code of Ethics. All datasets are either publicly available or curated by the authors with human validation to reduce bias and harmful content. No sensitive or personal data were used, and the methods are intended solely for scientific research.

## 8 REPRODUCIBILITY STATEMENT

We have made every effort to ensure the reproducibility of our work. All code and instructions necessary to reproduce the experiments in this paper are released on GitHub. The repository includes the full pipeline for dataset construction, xVerify training, and evaluation procedures in the paper. Detailed experimental settings and hyperparameters are documented in the main text and appendix, while additional implementation details are provided in the supplementary materials. Together, these resources allow for end-to-end reproduction of the results reported in this study.

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

# Appendices

# A USE OF LLM

In this work, LLMs were employed in several ways:

- Dataset construction and annotation: LLMs were used to assist in generating candidate data samples and providing preliminary annotations, which were subsequently verified and curated by human annotators.
- Experiments: LLMs were included as baselines or components in certain experiments, with details provided in the main text and appendix.
- Writing: LLMs were only used for auxiliary purposes such as improving clarity, grammar, and phrasing. They did not contribute to research ideation, methodological design, or substantive scientific writing.

All final decisions regarding data curation, experimental design, analysis, and writing were made by the authors, who take full responsibility for the content of the paper.

# B DATASETS AND MODELS

This section will present the relevant information for all the public datasets and LLMs involved in the experiments of this paper.

In this study, we employ a total of 24 datasets, which are categorized into four primary types: multiple-choice questions (Choice), short answer questions (Short Answer), mathematical problems (Math), and classification tasks (Classification), as summarized in Table 4. To evaluate the multilingual capabilities of the xVerify model, each question type includes datasets in both Chinese and English, with one dataset featuring multilingual content. For each dataset, samples are partitioned into training and test sets following a 2:1 ratio, with the training and test sets ideally comprising 2,000 and 1,000 instances, respectively. In certain cases, the number of available samples is below 3,000, or the official test set is not publicly available, resulting in reduced dataset sizes after preprocessing.

A total of 19 large language models (LLMs) are utilized in our experiments, encompassing a diverse range of model sizes and types, with a particular emphasis on reasoning models (see Table 5). These models are subsequently used to collect LLM-generated responses and to train the xVerify model.

# C VAR DATASET DETAILS

This section will present detailed information about the components of the VAR dataset, the details of human annotations, and examples from the dataset.

## C.1 DETAILS OF TRAINING, TEST, AND GENERALIZATION SETS

### C.1.1 TRAINING SET

The training set comprises 43,204 samples. Tables 6 to 9 provide the sample counts corresponding to each LLM, dataset, prompt template, and question type. Note that datasets with names containing "_enh" refer to the augmented multiple choice question datasets.

### C.1.2 TEST SET

The test set comprises 6,122 samples. Tables 10 to 13 provide the sample counts corresponding to each LLM, dataset, prompt template, and question type. Note that datasets with names containing "_enh" refer to the augmented multiple choice question datasets.

### C.1.3 GENERALIZATION SET

The generalization set comprises 6,468 samples. Tables 14 to 17 provide the sample counts corresponding to each LLM, dataset, prompt template, and question type. Note that datasets with names containing "_enh" refer to the augmented multiple choice question datasets.

Table 4: Datasets Description. The "Type" column indicates the question type in the corresponding dataset, including multiple-choice questions (Choice), short answer questions (Short Answer), math questions (Math), and classification questions (Classification).

| Dataset | Type | #Train | #Test | Language | License |
|---------|------|--------|-------|----------|---------|
| CMMLU | Choice | 2000 | 1000 | Chinese | CC-BY-NC-4.0 |
| C-Eval | Choice | 1346 | 260 | Chinese | CC-BY-NC-SA-4.0 |
| GPQA | Choice | 794 | 398 | English | CC-BY-4.0 |
| MMLU | Choice | 1816 | 1000 | English | MIT |
| MMLU-Pro | Choice | 2000 | 1000 | English | MIT |
| MMLU-Redux | Choice | 2000 | 1000 | English | CC-BY-4.0 |
| AgNews | Classification | 2000 | 1000 | English | Unspecified |
| Amazon | Classification | 2000 | 1000 | English | Apache-2.0 |
| CLUEWSC | Classification | 1548 | 1000 | Chinese | Unspecified |
| CMNLI | Classification | 2000 | 1000 | Chinese | Apache-2.0 |
| AMC23 | Math | 26 | 14 | English | Unspecified |
| AIME 2024 | Math | 20 | 10 | English | MIT |
| CMATH | Math | 1128 | 565 | Chinese | CC-BY-4.0 |
| GSM8K | Math | 2000 | 1000 | English | MIT |
| LiveMathBench | Math | 190 | 93 | English & Chinese | CC-BY-4.0 |
| MATH | Math | 2000 | 1000 | English | MIT |
| MGSM | Math | 1892 | 946 | Multilingual | CC-BY-SA-4.0 |
| OlympiadBench | Math | 1787 | 892 | English & Chinese | Apache-2.0 |
| ARC | Short Answer | 2000 | 1000 | English | CC-BY-SA-4.0 |
| CHID | Short Answer | 2000 | 1000 | Chinese | Apache-2.0 |
| C-SimpleQA | Short Answer | 2000 | 1000 | Chinese | CC-BY-NC-SA-4.0 |
| DROP | Short Answer | 2000 | 1000 | English | CC-BY-SA-4.0 |
| FRAMES | Short Answer | 550 | 274 | English | Apache-2.0 |
| SimpleQA | Short Answer | 2000 | 1000 | English | MIT |

Table 5: LLMs Description. LLMs are listed by release date. All models are chat or instruct type. "NaN" indicates that public data is unavailable.

| Model | #Para. | Type | Publisher | Date |
|-------|--------|------|-----------|------|
| ChatGLM3-6B | 6B | Chat | Tsinghua | 2023.10 |
| GPT-4o | NaN | Chat | OpenAI | 2024.05 |
| Gemma-2-2B-it | 2B | Instruct | Google | 2024.06 |
| Gemma-2-9B-it | 9B | Instruct | Google | 2024.06 |
| GLM-4-9B-Chat | 9B | Chat | Tsinghua | 2024.06 |
| InternLM2.5-7B-Chat | 7B | Chat | ShLab | 2024.06 |
| Qwen2-1.5B-Instruct | 1.5B | Instruct | Alibaba | 2024.06 |
| Qwen2-7B-Instruct | 7B | Instruct | Alibaba | 2024.06 |
| Llama-3.1-8B-Instruct | 8B | Instruct | Meta | 2024.07 |
| Llama-3.2-1B-Instruct | 1B | Instruct | Meta | 2024.09 |
| Llama-3.2-3B-Instruct | 3B | Instruct | Meta | 2024.09 |
| Qwen2.5-7B-Instruct | 7B | Instruct | Alibaba | 2024.09 |
| Qwen2.5-14B-Instruct | 14B | Instruct | Alibaba | 2024.09 |
| Phi-4 | 14B | Chat | Microsoft | 2024.11 |
| DeepSeek-R1-Distill-Llama-8B | 8B | Distill | DeepSeek | 2025.01 |
| DeepSeek-R1-Distill-Qwen-1.5B | 1.5B | Distill | DeepSeek | 2025.01 |
| DeepSeek-R1-Distill-Qwen-7B | 7B | Distill | DeepSeek | 2025.01 |
| DeepSeek-R1-Distill-Qwen-14B | 14B | Distill | DeepSeek | 2025.01 |
| QwQ-32B | 32B | Instruct | Alibaba | 2025.03 |

Table 6: Number of samples from each LLM in the training set.

| Model | Sample Counts |
|---|---|
| ChatGLM3-6B | 2588 |
| GPT-4o | 2691 |
| Gemma-2-2B-it | 2657 |
| Gemma-2-9B-it | 2600 |
| GLM-4-9B-Chat | 2957 |
| InternLM2.5-7B-Chat | 2935 |
| Qwen2-1.5B-Instruct | 2700 |
| Qwen2-7B-Instruct | 2898 |
| LLaMA-3.1-8B-Instruct | 2852 |
| Qwen2.5-7B-Instruct | 2854 |
| Qwen2.5-14B-Instruct | 2801 |
| DeepSeek-R1-Distill-Llama-8B | 3223 |
| DeepSeek-R1-Distill-Qwen-1.5B | 3231 |
| DeepSeek-R1-Distill-Qwen-7B | 3075 |
| DeepSeek-R1-Distill-Qwen-14B | 3142 |

Table 7: Number of samples from each dataset in the training set.

| Dataset | Sample Counts |
|---|---|
| CMMLU | 1557 |
| CMMLU_enh | 1641 |
| GPQA | 1587 |
| GPQA_enh | 1668 |
| MMLU | 1520 |
| MMLU_enh | 1513 |
| MMLU-Pro | 1394 |
| MMLU-Pro_enh | 1442 |
| AgNews | 1751 |
| CLUEWSC | 5008 |
| AMC23 | 1625 |
| AIME 2024 | 1333 |
| CMATH | 1893 |
| GSM8K | 1836 |
| MATH | 2485 |
| MGSM | 1384 |
| OlympiadBench_en | 2573 |
| OlympiadBench_zh | 2709 |
| CHID | 2424 |
| C-SimpleQA | 1913 |
| DROP | 1928 |
| FRAMES | 2020 |

Table 8: Number of samples from each prompt template in the training set.

| Prompt Template | Sample Counts |
|---|---|
| 0-shot | 4884 |
| 0-shot-restrict | 5977 |
| 0-shot-cot | 4907 |
| 0-shot-cot-restrict | 6041 |
| 5-shot | 4774 |
| 5-shot-restrict | 5866 |
| 5-shot-cot | 4916 |
| 5-shot-cot-restrict | 5839 |

Table 9: Number of samples from each question type in the training set.

| Dataset | Sample Counts |
|---|---|
| Multiple Choice | 12322 |
| Math | 15838 |
| Short Answer | 8285 |
| Classification | 6759 |

Table 10: Number of samples from each LLM in the test set.

| Model | Sample Counts |
|---|---|
| ChatGLM3-6B | 378 |
| GPT-4o | 400 |
| Gemma-2-2B-it | 416 |
| Gemma-2-9B-it | 369 |
| GLM-4-9B-Chat | 367 |
| InternLM2.5-7B-Chat | 367 |
| Qwen2-1.5B-Instruct | 433 |
| Qwen2-7B-Instruct | 427 |
| LLaMA-3.1-8B-Instruct | 404 |
| Qwen2.5-7B-Instruct | 374 |
| Qwen2.5-14B-Instruct | 415 |
| DeepSeek-R1-Distill-Llama-8B | 430 |
| DeepSeek-R1-Distill-Qwen-1.5B | 451 |
| DeepSeek-R1-Distill-Qwen-7B | 439 |
| DeepSeek-R1-Distill-Qwen-14B | 452 |

Table 11: Number of samples from each dataset in the test set.

| Dataset | Sample Counts |
|---|---|
| CMMLU | 216 |
| CMMLU_enh | 195 |
| GPQA | 207 |
| GPQA_enh | 235 |
| MMLU | 225 |
| MMLU_enh | 222 |
| MMLU-Pro | 171 |
| MMLU-Pro_enh | 192 |
| AgNews | 261 |
| CLUEWSC | 710 |
| AMC23 | 258 |
| AIME 2024 | 186 |
| CMATH | 263 |
| GSM8K | 262 |
| MATH | 362 |
| MGSM | 205 |
| OlympiadBench_en | 349 |
| OlympiadBench_zh | 446 |
| CHID | 347 |
| C-SimpleQA | 270 |
| DROP | 265 |
| FRAMES | 275 |

Table 12: Number of samples from each prompt template in the test set.

| Dataset | Sample Counts |
|---|---|
| Multiple Choice | 1663 |
| Math | 2331 |
| Short Answer | 1157 |
| Classification | 971 |

Table 13: Number of samples from each question type in the test set.

| Prompt Template | Sample Counts |
|---|---|
| 0-shot | 680 |
| 0-shot-restrict | 798 |
| 0-shot-cot | 642 |
| 0-shot-cot-restrict | 891 |
| 5-shot | 690 |
| 5-shot-restrict | 789 |
| 5-shot-cot | 702 |
| 5-shot-cot-restrict | 930 |

Table 14: Number of samples from each LLM in the generalization set.

| Model | Sample Counts |
|---|---|
| ChatGLM3-6B | 300 |
| GPT-4o | 305 |
| Gemma-2-2B-it | 427 |
| Gemma-2-9B-it | 296 |
| GLM-4-9B-Chat | 339 |
| InternLM2.5-7B-Chat | 341 |
| Qwen2-1.5B-Instruct | 280 |
| Qwen2-7B-Instruct | 346 |
| LLaMA-3.1-8B-Instruct | 400 |
| LLaMA-3.2-1B-Instruct | 314 |
| LLaMA-3.2-3B-Instruct | 310 |
| Qwen2.5-7B-Instruct | 326 |
| Qwen2.5-14B-Instruct | 334 |
| Phi-4 | 314 |
| DeepSeek-R1-Distill-Llama-8B | 341 |
| DeepSeek-R1-Distill-Qwen-1.5B | 399 |
| DeepSeek-R1-Distill-Qwen-7B | 375 |
| DeepSeek-R1-Distill-Qwen-14B | 434 |
| QwQ-32B | 287 |

Table 15: Number of samples from each dataset in the generalization set.

| Dataset | Sample Counts |
|---|---|
| C-Eval | 435 |
| C-Eval_enh | 442 |
| MMLU-Redux | 436 |
| MMLU-Redux_enh | 483 |
| Amazon | 646 |
| CMNLI | 643 |
| LiveMathBench_en | 1127 |
| LiveMathBench_zh | 821 |
| ARC | 807 |
| SimpleQA | 628 |

Table 16: Number of samples from each prompt template in the generalization set.

| Dataset | Sample Counts |
| --- | --- |
| Multiple Choice | 1796 |
| Math | 1948 |
| Short Answer | 1435 |
| Classification | 1289 |

Table 17: Number of samples from each question type in the generalization set.

| Prompt Template | Sample Counts |
| --- | --- |
| 0-shot | 703 |
| 0-shot-restrict | 856 |
| 0-shot-cot | 772 |
| 0-shot-cot-restrict | 915 |
| 5-shot | 690 |
| 5-shot-restrict | 885 |
| 5-shot-cot | 756 |
| 5-shot-cot-restrict | 891 |

## C.2   DETAILS OF HUMAN ANNOTATION

To ensure high-quality annotation for the VAR dataset, we assembled a team of 8 annotators. Among them, 6 hold bachelor's degrees and are primarily responsible for batch annotation tasks, while the other 2 hold master's degrees and focus on reviewing complex cases or resolving discrepancies in annotations made by multiple annotators. The gender ratio within the annotation team is balanced at 1:1. In terms of compensation, all annotators were paid according to the local industry average rates. The annotation process lasted for three weeks, covering a total of 15 working days.

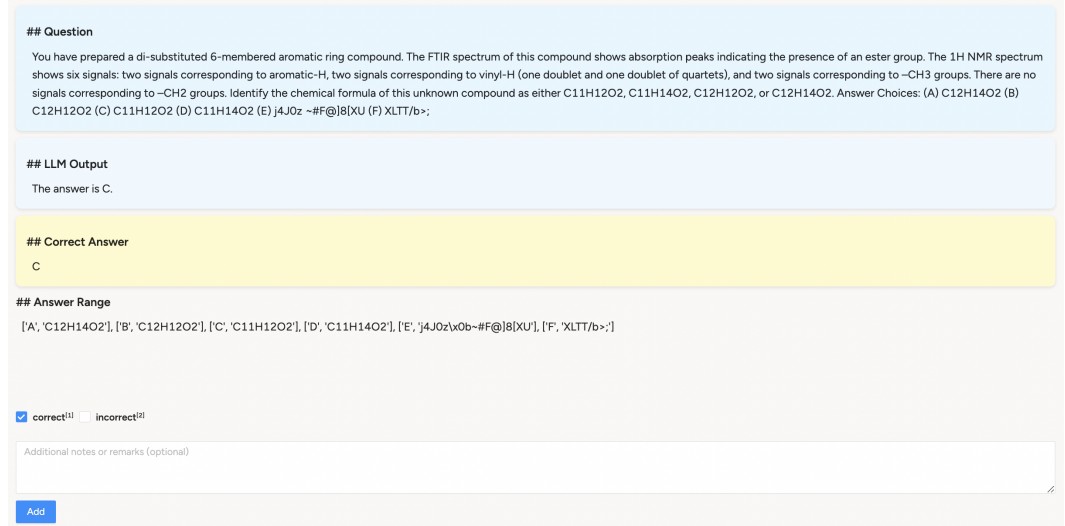

Figure 3: Illustration of the Label Studio Interface.

The detailed annotation guidelines are presented below. Figure 3 shows an example of the interface used in our annotation tool. Each sample to be annotated contains four fields: `question`, `LLM output`, `correct answer`, and `answer range`. The `question type` includes four categories: *multiple choice*, *math*, *short answer*, and *classification*. Annotators are required to judge whether the `LLM output` matches the `correct answer` based on the `question`, while the

`answer range` serves as auxiliary reference information to support the decision-making process. The specific annotation instructions and criteria are as follows:

**Answer evaluation criteria for different question types:**

- **Multiple Choice**
  For multiple-choice questions, answer options may be labeled with letters (A, B, C, D, . . . ), Roman numerals (I, II, III, IV, . . . ), or Arabic numerals (1, 2, 3, 4, . . . ). The LLM output is considered *correct* if it provides:
    - Only the correct option label;
    - Only the correct option content;
    - Both the correct label and content.

  In cases where the label and content are inconsistent, the **content takes precedence**. If the content is correct, the answer is marked as *correct*; if the content is incorrect, the answer is marked as *incorrect*, even if the option label is correct (see the final annotation example for reference).

- **Short Answer**
  Short-answer questions may require responses such as names, locations, numbers, dates, or full sentences. The evaluation criteria are:
    - For concise answers (e.g., names, places, dates), strict string matching is required.
    - For sentence-level answers, semantic consistency with the reference answer is required.
    - For numerical answers, mathematical equivalence must be verified (e.g., "12000" and "12,000" are considered equivalent).

- **Classification**
  Classification questions come with a fixed set of candidate answers. The LLM output must explicitly and exactly match the correct answer in this set to be judged as *correct*.

- **Math**
  For mathematical questions, the final answer in the LLM output must be mathematically equivalent to the reference answer. Evaluation criteria include:
    - If an initial answer (`ans1`) is given but followed by a derived final answer (`ans2`) through calculation, `ans2` should be used for evaluation.
    - If the LLM output or ground-truth answer is provided in LaTeX format and cannot be visually interpreted, a LaTeX compiler should be used to determine equivalence.

**Special cases:**

- **Overly Long Responses**
  If the LLM output is excessively long, use the final answer provided as the basis for judgment. If the response does not converge to a clear answer (e.g., repeated changes or ambiguity), it should be marked as *incorrect*.

- **Truncated Calculations**
  In long responses where the final verification or calculation is truncated, it can be ignored. If a clear answer was provided earlier, use it for evaluation; if not, mark the response as *incorrect* (see the second-to-last annotation example).

- **Unfamiliar Domains**
  If the correctness of the LLM response cannot be determined due to unfamiliar domain knowledge (e.g., mathematical expressions whose equivalence is hard to judge), the case may be skipped and will later be annotated by more qualified annotators.

- **Irrelevant but Matched Answers**
  If the LLM output is irrelevant to the question but coincidentally contains the correct final answer, it should be marked as *incorrect*.

## C.3 EXAMPLES FROM THE VAR DATASET

```
{
    "dataset": "SimpleQA",
    "question": "In which year did Fayaz A. Malik (an Indian
    pharmacologist, cancer biologist, and scientist) receive the Young
    Scientist of the Year from the Council of Scientific and Industrial
    Research?",
    "question_type": "short answer",
    "correct_answer": "2009",
    "llm_output": "The year Fayaz A. Malik received the Young Scientist
    of the Year award from the Council of Scientific and Industrial
    Research was 2001.\textbackslash n\textbackslash nThe answer is
    2001.",
    "human_judgment_result": "Incorrect"
},
{
    "dataset": "MMLU-Redux_enh",
    "question": "If you wanted to find the global distribution of coal,
    you would use a  Answer Choices:  (I) reference map.  (II)
    topographic map.  (III) thematic map.  (IV) location map.",
    "question_type": "multiple choice",
    "correct_answer": "III",
    "llm_output": "The answer is (III) thematic map.",
    "human_judgment_result": "Correct"
},
{
    "dataset": "Amazon",
    "question": "Lightsaber Action!: This game is absolutly the best game
     I have ever seen! there is a little bit of violence, but most of the
     time you kill droids, not humans. My favorite weopon is the
    lightsaber, because you can deflect laser blasts.\textbackslash n
    Please identify the sentiment polarity of the sentence: positive or
    negative",
    "question_type": "classification",
    "correct_answer": "positive",
    "llm_output": "The answer is positive.",
    "human_judgment_result": "Correct"
},
{
    "dataset": "GSM8K",
    "question": "Daisy and Rose were enjoying their backyard pool with
    their dogs.  If there are 24 legs/paws in the pool, how many dogs do
    Daisy and Rose have?",
    "question_type": "math",
    "correct_answer": "5",
    "llm_output": "To find the number of dogs Daisy and Rose have, we
    need to first determine the total number of people and then subtract
    that from the total number of legs/paws.\n\nSince each person has 2
    legs, we can divide the total number of legs/paws by 2 to find the
    number of people:\n24 legs/paws / 2 legs/person = 12 people\n\nSince
    Daisy and Rose are the only people in the pool, we can subtract them
    from the total number of people to find the number of dogs:\n12
    people - 2 people = 10 dogs\n\nTherefore, Daisy and Rose have 10 dogs
    .",
    "human_judgment_result": "Incorrect"
}
```

## D  MODEL TRAINING DETAILS

This section will further present additional information about the training of the xVerify model.

### D.1  TRAINING HYPERPARAMETERS

The xVerify model is trained using the QLoRA method, with consistent hyperparameter settings across all base models. The training is carried out on multiple GPU servers. Table 18 presents the key training hyperparameters.

Table 18: Hyperparameter settings for model training.

| Hyperparameter | Setting |
|---|---|
| Per Device Train Batch Size | 1 |
| Gradient Accumulation Steps | 8 |
| Learning Rate | 1.0e-4 |
| Num Train Epochs | 1.0 |
| LrScheduler Type | cosine |
| Warmup Ratio | 0.1 |
| Bf16 | true |
| Ddp Timeout | 180000000 |
| Lora Rank | 8 |

### D.2  ORIGINAL MODEL DETAILS

This paper uses 14 original models of different parameter scales and types for training on the VAR dataset. Table 19 presents the relevant information for all xVerify models and their corresponding original models.

Table 19: Details of Original Models and Corresponding xVerify Models. Sorted by Original Model Name.

| Original Model | #Para. | Type | Context Length | xVerify Model |
|---|---|---|---|---|
| Gemma-2-2B-it | 2B | Instruct | 8K | xVerify-2B-I |
| Gemma-2-9B-it | 9B | Instruct | 8K | xVerify-9B-I |
| Gemma-2-27B-it | 27B | Instruct | 8K | xVerify-27B-I |
| GLM-4-9B-Chat | 9B | Chat | 128K | xVerify-9B-C |
| Llama-3.2-1B-Instruct | 1B | Instruct | 128K | xVerify-1B-I |
| Llama-3.2-3B-Instruct | 3B | Instruct | 128K | xVerify-3B-Ia |
| Llama-3.1-8B-Instruct | 8B | Instruct | 128K | xVerify-8B-I |
| Phi-4 | 14B | Instruct | 16k | xVerify-14B-Ib |
| Qwen2.5-0.5B-Instruct | 0.5B | Instruct | 128K | xVerify-0.5B-I |
| Qwen2.5-1.5B-Instruct | 1.5B | Instruct | 128K | xVerify-1.5B-I |
| Qwen2.5-3B-Instruct | 3B | Instruct | 128K | xVerify-3B-Ib |
| Qwen2.5-7B-Instruct | 7B | Instruct | 128K | xVerify-7B-I |
| Qwen2.5-14B-Instruct | 14B | Instruct | 128K | xVerify-14B-Ia |
| Qwen2.5-32B-Instruct | 32B | Instruct | 128K | xVerify-32B-I |

## E  RL TRAINING DETAILS

This section will further present additional information about RL training.

### E.1 RL TRAINING HYPERPARAMETERS

We train our models using the veRL (Sheng et al., 2024) Framework. The hyperparameters for implementing RL are presented in Table 20. When implementing baseline methods, we use the same hyperparameters. The training is carried out on multiple GPU servers.

Table 20: Hyperparameter settings for RL training.

| Hyperparameter | Setting |
|---|---|
| Epoch | 15 |
| Learning Rate | 1.0e-6 |
| Train Batch Size | 1024 |
| Maximum Rollout Length | 8192 |
| KL Loss Coefficient | 1e-3 |
| Entropy Coefficient | 0.0 |
| Temperature | 1.0 |
| Rollouts per Prompt | 5 |

### E.2 DETAILS OF RL TRAINING AND GENERALIZATION SETS

When training policy, we use xVerify as the reward model. To ensure that the evaluation truly reflects the generalization capability of xVerify, we select all training data from the generalization set, thereby avoiding data contamination. The training set comprises 20,400 samples from different types of questions as shown in Table 21. The test set comprises 1,000 samples as shown in Table 22.

Table 21: Number of samples from each dataset in RL training set.

| Dataset | Sample Counts |
|---|---|
| MMLU-Pro | 3400 |
| MATH | 3400 |
| DROP | 3400 |
| GSM8K | 3400 |
| AgNews | 3400 |
| ARC | 3400 |

### E.3 DETAILS OF RL TRAINING AND GENERALIZATION SETS

The training set comprises 20,400 samples. The specific composition is shown in Table 21. The generalization set comprises 1,750 samples as shown in Table 22.

Table 22: Number of samples from each dataset in RL generalization set.

| Dataset | Sample Counts |
|---|---|
| MATH-500 | 470 |
| MMLU-Pro | 250 |
| DROP | 250 |
| GPQA | 250 |
| Amazon | 250 |
| CMMLU | 250 |
| CHID | 250 |

# F  PROMPTS

This section will present all the prompt templates used in the experiments of this paper.

## F.1  PROMPTS FOR GENERATING LLM RESPONSES

The prompt templates used to generate LLM responses are illustrated in Figures 4 to 7. Each template consists of four fields that need to be populated: "task_type", "task_description", "examples", and "question". The "task_type" and "task_description" fields are determined based on the type of question. For instance, for questions from the GPQA dataset, "task_type" is set to "multidisciplinary question", and "task_description" is set to "Please choose the answer from options A to D, corresponding to the question." During dataset preprocessing, we design appropriate "task_type" and "task_description" values for each dataset. The "examples" field is filled according to the selected prompting strategy, either 0-shot or 5-shot. In the 0-shot setting, this field is left empty, while in the 5-shot setting, it is populated with five example question-answer pairs that are similar to the target "question". The "question" field contains the specific query to be answered by the LLM. Examples of the "examples" and "question" fields are shown in Figures 8 and 9, respectively.

You are an expert in {task_type}, {task_description}
{examples}
{question}

Figure 4: Few-shot prompt for generating LLM responses.

You are an expert in {task_type}, {task_description}
{examples}
{question}

End your final answer with 'The answer is <answer>.'

Figure 5: Few-shot-restrict prompt for generating LLM responses.

You are an expert in {task_type}, {task_description}
{examples}
{question}

Let's think step by step.

Figure 6: Few-shot-cot prompt for generating LLM responses.

## F.2  PROMPTS FOR GPT-4O ANNOTATION

The prompt templates used for annotating the collected LLM question-answer pairs with GPT-4o during the construction of the VAR dataset are shown in Figures 10 and 11. Both of these prompt templates employ the Chain-of-Thought (CoT) strategy to ensure the accuracy of the annotations generated by GPT-4o.

## F.3  PROMPTS FOR DATA AUGMENTATION

In constructing the VAR dataset, two prompt templates used to guide GPT-4o in augmenting mathematical question samples are presented in Figures 12 and 13.

You are an expert in {task_type}, {task_description}
{examples}
{question}

Let's think step by step.

End your final answer with 'The answer is <answer>.'

Figure 7: Few-shot-cot-restrict prompt for generating LLM responses.

***** Start In-Context Examples *****
Q: A late game rally by Washington led them to the Eagles' 26 yard line. A shot to the end zone by Robert Griffin III would be intercepted by Brandon Boykin, clinching an Eagles win. The Eagles would move to 6-5. This is the Eagles first win at Lincoln Financial Field since Week 4 of the 2012 season, because prior to this game, the Eagles had never won a game in their home stadium in 414 days since that same week, snapping a 10-game losing streak at home with this win. How many more wins than losses did the Eagles have after this game?
A: The answer is 1.

Q: The population of Sevastopol proper is 418,987 (01.01.16), making it the largest in the Crimean Peninsula. The citys agglomeration has about 600,000 people (2015). According to the Ukrainian Census (2001), the ethnic groups of Sevastopol include Russians (71.6%), Ukrainians (22.4%), Belarusians (1.6%), Tatars (0.7%), Crimean Tatars (0.5%), Armenians (0.3%), Jews (0.3%), Moldovans (0.2%), and Azerbaijani people (0.2%). Which ethnic has a higher percentage of the population in Sevastopol: Russians or Armenians?
A: The answer is Russians.

Q: the most common crimes in the ACT are property related crimes, unlawful entry with intent and motor vehicle theft. They affected 2,304 and 966 people (580 and 243 per 100,000 persons respectively). Homicide and related offences—murder, attempted murder and manslaughter, but excluding driving causing death and conspiracy to murder—affect 1.0 per 100,000 persons, which is below the national average of 1.9 per 100,000. Rates of sexual assault (64.4 per 100,000 persons) are also below the national average (98.5 per 100,000). Which was there a higher national average for, homicide and related offences or sexual assault?
A: The answer is sexual assault.

Q: In the county, the population was spread out with 21.7% under the age of 18, 8.5% from 18 to 24, 26.9% from 25 to 44, 27.7% from 45 to 64, and 15.0% who were 65 years of age or older. The median age was 40 years. For every 100 females, there were 94.4 males. For every 100 females age 18 and over, there were 98.7 males. How many percent were not from 45 to 64?
A: The answer is 72.3.

Q: The median age in the city was 35.1 years. 24.2% of residents were under the age of 18; 7.9% were between the ages of 18 and 24; 33.8% were from 25 to 44; 24.6% were from 45 to 64; and 9.5% were 65 years of age or older. The gender makeup of the city was 48.6% male and 51.4% females. How many more people, in terms of percentage, were in the largest age group compared to the second smallest?
A: The answer is 24.3.
***** End In-Context Examples *****

Figure 8: Example of "examples" fields.

Q: Let $ABCD$ be a tetrahedron such that $AB = CD = \sqrt{41}$, $AC = BD = \sqrt{80}$, and $BC = AD = \sqrt{89}$. There exists a point $I$ inside the tetrahedron such that the distances from $I$ to each of the faces of the tetrahedron are all equal. This distance can be written in the form $\frac{m\sqrt{n}}{p}$, where $m$, $n$, and $p$ are positive integers, $m$ and $p$ are relatively prime, and $n$ is not divisible by the square of any prime. Find $m + n + p$.
A:

Figure 9: Example of "question" fields.

You are a diligent and precise assistant tasked with evaluating the correctness of responses. Think step by step as you make your evaluation.

You will receive a question, an output sentence, and the correct answer. Your task is to determine if the output sentence accurately answers the question based on the provided correct answer. Think step by step and respond with either [Correct] or [Incorrect].
-
Special considerations:
1. **Multiple Answers**: If the output contains multiple answers, evaluate whether later answers modify or correct earlier ones. In such cases, compare the final answer with the correct answer. If the final answer is unclear or incorrect, respond with [Incorrect].
2. **Mathematical Problems**: If the formats differ but the answers are mathematically equivalent, respond with [Correct].
3. **Explicit Options**: If the question provides explicit candidate answers, the output will be considered correct if it clearly indicates the correct option's code or the correct option's content.
4. **No Explicit Options**: If the question does not provide explicit options, the output must align with the correct answer in content and meaning to be considered [Correct].
Please present your response in the following JSON format:
{
        "reasoning": "Your step-by-step reasoning here.",
        "judgment": "Correct or Incorrect"
}
-
Question: """{question}"""
Output sentence: """{output}"""
Correct answer: {answer}

Figure 10: Prompt I for GPT-4o annotation.

> You are a diligent and precise assistant tasked with evaluating the correctness of responses. Think step by step as you make your evaluation.
>
> We request your feedback on whether the model's response correctly answers the user question above. Follow these steps to make your evaluation:
> 1. Think step by step: Read the user question carefully.
> 2. Think step by step: Review the reference answer and understand the key points it covers.
> 3. Think step by step: Compare the model's answer with the reference answer.
> 4. Think step by step: Determine if the model's answer addresses the key points in the reference answer and correctly answers the question.
> -
> First, provide your reasoning in detail. Then, clearly state your judgment as either "Correct" or "Incorrect."
> Please present your response in the following JSON format:
> {
>     "reasoning": "Your step-by-step reasoning here.",
>     "judgment": "Correct or Incorrect"
> }
> -
> Question: {question}
> Reference Answer: {answer}
> Model's Answer: {output}

Figure 11: Prompt II for GPT-4o annotation.

### F.4 PROMPTS FOR JUDGE MODEL

In the experiments of this paper, the prompts used for all judge models were constructed based on the official templates provided by their respective developers. However, for some judge models, the official prompt templates were not fully compatible with the evaluation tasks in this paper, so other similar prompt templates were used. Specifically, Figure 14 shows the prompt template used by GPT-4o as Judge, Figure 15 shows the prompt template used by GPT-4o as Judge (CoT), Figure 16 shows the prompt template used by JudgeLM series models and PandaLM-7B-v1, Figure 17 shows the prompt template used by Auto-J series models, and Figure 18 shows the prompt template used by Prometheus series models. The official prompt template for the CompassJudger-1 series models corresponds to pairwise evaluation, so the prompt template used by this series is the same as that for the xVerify model, as shown in Figure 19.

### F.5 PROMPTS FOR XVERIFY

Figure 19 shows the prompt template used to construct the input for the xVerify model. This template is used both for training and evaluation of the xVerify model. Specifically, "question," "output," and "answer" correspond to the question content, the LLM response, and the reference answer, respectively.

## G SUPPLEMENTARY EXPERIMENTAL RESULTS

### G.1 EVALUATION ACCURACY RESULTS OF ALL XVERIFY MODELS AND JUDGE MODELS

Tables 23 and 24 present the performance of all xVerify models and judge models on the test set and generalization set, respectively. Overall, each xVerify model achieves an F1 score and accuracy exceeding 96.5% on the test set, and an accuracy greater than 95.52% on the generalization set. These results not only demonstrate the effectiveness of xVerify models in the evaluation task—consistently outperforming all other judge models—but also underscore the high quality of the VAR dataset.

A comparison of the results across the two datasets reveals that the performance of xVerify models on the generalization set exhibits a slight decline relative to the test set, with a maximum drop of no more than 1.6%. Moreover, xVerify models with larger parameter sizes tend to show a smaller per-

```
You are an expert in mathematical calculations and data expressions.
You are required to provide different equivalent forms of the standard
answer for the following math problem.
Problem: {question}
Answer: {answer}

Example 1:
Problem: Let $\alpha$ be the radian measure of the smallest angle in a
$3-4-5$ right triangle. Let $\beta$ be the radian measure of the
smallest angle in a $7-24-25$ right triangle. Express $\beta$ in terms
of $\alpha$.
Answer: $\frac{\pi}{2} - 2\alpha$
Output:
```json {
    "answer1": "\pi/2 - 2\alpha",
    "answer2": "pi/2 - 2alpha",
    "answer3": "pi/2 - 2 * alpha",
    "answer4": "0.5 * pi - 2 * alpha" }```

Example 2:
Problem: A volcano erupts and spews ash into the sky. The ash cloud
spreads out in a diameter eighteen times as far as the distance it
shot up into the sky. If the ashes erupted three hundred feet into the
sky, what was the radius of the ash cloud in feet?
Answer: 2700
Output:
```json {
    "answer1": "2.7×10^3",
    "answer2": "2700.0",
    "answer3": "2.7 \times 10^3",
    "answer4": "$2.7 \times 10^3$",
    "answer5": "Two thousand seven hundred" }```

Please note:
1. You need to provide 3 to 5 different standard forms of the answer.
2. Each different form must be equivalent to the standard answer, i.e.,
it should still be a correct and valid answer.
3. You may use LaTeX, scientific notation, or other standard
mathematical expressions.
4. Please follow the JSON format below for the output:
```json {
    "answer1": "xxx", "answer2": "xxx", "answer3": "xxx", ...
}```
```

Figure 12: Prompt for Generating Alternative Reference Answers.

```
You are an expert in mathematical calculations and data expressions.
For an answer to a specific mathematical problem, you are required to
provide equivalent and different expressions of the mathematical
result.
Answer: {output}

Example 1:
Answer: The answer is $\beta = \frac{\pi}{2} - 2\alpha$.
Output:
```json {
    "answer1": "The answer is \pi/2 - 2\alpha.",
    "answer2": "The answer is pi/2 - 2alpha.",
    "answer3": "The answer is pi/2 - 2 * alpha.",
    "answer4": "The answer is 0.5 * pi - 2 * alpha."
}```

Example 2:
Answer: The answer is 2700 feet.
Output:
```json {
    "answer1": "The answer is 2.7×10^3 feet.",
    "answer2": "The answer is 2700.0 feet.",
    "answer3": "The answer is 2.7 \times 10^3 feet.",
    "answer4": "The answer is $2.7 \times 10^{3}$ feet.",
    "answer5": "The answer is Two thousand seven hundred feet."
}```

Please note:
1. You need to provide 3 to 5 different expressions, each replacing
the mathematical result with an equivalent and different form.
2. Each expression must be exactly equivalent to the target answer to
ensure its correctness.
3. You can use LaTeX, scientific notation, or other standard
mathematical formats.
4. Please output the result in the following JSON format:
```json {
    "answer1": "The answer is xxx",
    "answer2": "The answer is xxx",
    "answer3": "The answer is xxx",
    "answer4": "The answer is xxx",
    "answer5": "The answer is xxx"
}```
```

Figure 13: Prompt for Generating Diverse Final Answer Expressions.

You are a diligent and precise assistant tasked with evaluating the correctness of responses. You will receive a question, an output sentence, and the correct answer. Your task is to determine if the output sentence accurately answers the question based on the provided correct answer. Respond with either [Correct] or [Incorrect].
-
Special considerations:
1. **Multiple Answers**: If the output contains multiple answers, evaluate whether later answers modify or correct earlier ones. In such cases, compare the final answer with the correct answer. If the final answer is unclear or incorrect, respond with [Incorrect].
2. **Mathematical Problems**: If the formats differ but the answers are mathematically equivalent, respond with [Correct].
3. **Explicit Options**: If the question provides explicit candidate answers, the output will be considered correct if it clearly indicates the correct option's code or the correct option's content.
4. **No Explicit Options**: If the question does not provide explicit options, the output must align with the correct answer in content and meaning to be considered [Correct].
Please present your response in the following JSON format:
{
        "judgement": "Correct or Incorrect"
}
-
Question: """{question}"""
Output sentence: """{response}"""
Correct answer: {reference}

Figure 14: Prompt for GPT-4o as Judge.

You are a diligent and precise assistant tasked with evaluating the correctness of responses. Think step by step as you make your evaluation. You will receive a question, an output sentence, and the correct answer. Your task is to determine if the output sentence accurately answers the question based on the provided correct answer. Think step by step and respond with either [Correct] or [Incorrect].
-
Special considerations:
1. **Multiple Answers**: If the output contains multiple answers, evaluate whether later answers modify or correct earlier ones. In such cases, compare the final answer with the correct answer. If the final answer is unclear or incorrect, respond with [Incorrect].
2. **Mathematical Problems**: If the formats differ but the answers are mathematically equivalent, respond with [Correct].
3. **Explicit Options**: If the question provides explicit candidate answers, the output will be considered correct if it clearly indicates the correct option's code or the correct option's content.
4. **No Explicit Options**: If the question does not provide explicit options, the output must align with the correct answer in content and meaning to be considered [Correct].
Please present your response in the following JSON format:
{
        "reasoning": "Your step-by-step reasoning here.",
        "judgement": "Correct or Incorrect"
}
-
Question: """{question}"""
Output sentence: """{response}"""
Correct answer: {reference}

Figure 15: Prompt for GPT-4o as Judge (CoT).

You are a helpful and precise assistant for checking the quality of the answer.
[Question]
{question}
[Reference Answer]
{reference}
[Model's Answer]
{response}
[System]
We would like to request your feedback on the performance of the model's response to the user question displayed above.
Based on the reference answer, please rate the accuracy of the response. The model receives an overall score on a scale of 1 to 10, where a higher score indicates better overall performance.
Please first output a single line containing only the score. In the subsequent line, please provide a comprehensive explanation of your evaluation, avoiding any potential bias.

### Response:

Figure 16: Prompt for JudgeLM.

[INST] Write critiques for a submitted response on a given user's query, incorporating the correct answer as a reference, and grade the response accordingly:

[BEGIN DATA]
***
[Query]: {question}
***
[Correct Answer]: {reference}
***
[Response]: {response}
***
[END DATA]

Write critiques for this response. After that, you should give a final rating for the response on a scale of 1 to 10 by strictly following this format: "[[rating]]", for example: "Rating: [[5]]".
[/INST]

Figure 17: Prompt for Auto-J.

You are a fair judge assistant tasked with providing clear, objective feedback based on specific criteria, ensuring each assessment reflects the absolute standards set for performance."
###Task Description:
An instruction (might include an Input inside it), a response to evaluate, a reference answer that gets a score of 5, and a score rubric representing a evaluation criteria are given.
1. Write a detailed feedback that assess the quality of the response strictly based on the given score rubric, not evaluating in general.
2. After writing a feedback, write a score that is an integer between 1 and 5. You should refer to the score rubric.
3. The output format should look as follows: "Feedback: (write a feedback for criteria) [RESULT] (an integer number between 1 and 5)" 4. Please do not generate any other opening, closing, and explanations.

###The instruction to evaluate:
{question}

###Response to evaluate:
{response}

###Reference Answer (Score 5):
{reference}

###Score Rubrics:
[Does the model demonstrate logical and effective reasoning in its responses?]
Score 1: The model's responses show a complete lack of logical reasoning, often resulting in irrelevant or nonsensical answers.
Score 2: The model occasionally shows signs of logical reasoning but generally struggles to provide coherent or relevant responses.
Score 3: The model usually demonstrates basic reasoning capabilities, though it may not consistently apply logical principles or fully resolve complex issues.
Score 4: The model frequently exhibits strong reasoning skills, effectively addressing complex questions with minor inconsistencies or errors.
Score 5: The model consistently demonstrates advanced reasoning abilities, providing logically sound, coherent, and sophisticated responses to complex queries.

###Feedback:

Figure 18: Prompt for Prometheus.

> You are a diligent and precise assistant tasked with evaluating the correctness of responses. You will receive a question, an output sentence, and the correct answer. Your task is to determine if the output sentence accurately answers the question based on the provided correct answer. Respond with either [Correct] or [Incorrect].
> -
> Special considerations:
> 1. **Multiple Answers**: If the output contains multiple answers, evaluate whether later answers modify or correct earlier ones. In such cases, compare the final answer with the correct answer. If the final answer is unclear or incorrect, respond with [Incorrect].
> 2. **Mathematical Problems**: If the formats differ but the answers are mathematically equivalent, respond with [Correct].
> 3. **Explicit Options**: If the question provides explicit candidate answers, the output will be considered correct if it clearly indicates the correct option's code or the correct option's content.
> 4. **No Explicit Options**: If the question does not provide explicit options, the output must align with the correct answer in content and meaning to be considered [Correct].
> -
> Question: """{question}"""
> Output sentence: """{output}"""
> Correct answer: {answer}
> Judgement:

Figure 19: Prompt for xVerify.

formance degradation, indicating strong generalization capabilities that further improve with model scale. Additionally, it is observed across both datasets that while the performance of xVerify models generally enhances with the increment of parameter size, beyond a certain threshold, further increases in parameter scale do not lead to additional performance gains.

Table 23: Evaluation Accuracy Results on the Test Set: All xVerify Models and Judge Models. The best performance in each column is shown in **bold**, and the second-best performance is underlined.

| Method Type | Method | Multiple Choice | | Math | | Short Answer | | Classification | | Overall | |
|---|---|---|---|---|---|---|---|---|---|---|---|
| | | F1 | Acc. | F1 | Acc. | F1 | Acc. | F1 | Acc. | F1 | Acc. |
| | PandaLM-7B-v1 | 4.26% | 8.12% | 16.78% | 14.46% | 23.47% | 17.72% | 25.32% | 16.79% | 16.40% | 13.72% |
| | Auto-J-Bilingual-6B | 52.85% | 67.71% | 40.76% | 65.21% | 67.22% | 79.60% | 74.86% | 71.37% | 57.04% | 69.59% |
| | Auto-J-13B | 40.00% | 63.20% | 26.32% | 60.62% | 64.41% | 78.22% | 86.04% | 82.60% | 53.38% | 68.13% |
| | Prometheus-7B-v2.0 | 75.76% | 75.41% | 74.20% | 74.35% | 70.95% | 74.59% | 84.80% | 77.03% | 76.50% | 75.11% |
| | Prometheus-8x7B-v2.0 | 71.26% | 68.61% | 71.99% | 66.92% | 76.24% | 77.70% | 83.27% | 77.65% | 74.57% | 71.12% |
| Judge | JudgeLM-7B-v1.0 | 56.53% | 42.57% | 46.09% | 34.58% | 60.33% | 50.56% | 83.89% | 73.22% | 59.02% | 45.90% |
| Model | JudgeLM-13B-v1.0 | 56.81% | 48.89% | 58.39% | 59.46% | 77.32% | 79.52% | 95.63% | 93.82% | 68.57% | 65.83% |
| | JudgeLM-33B-v1.0 | 42.86% | 43.24% | 44.82% | 46.03% | 57.86% | 62.23% | 73.42% | 67.56% | 52.00% | 51.75% |
| | CompassJudger-1-1.5B | 49.95% | 35.54% | 61.66% | 48.78% | 57.36% | 46.93% | 82.51% | 70.96% | 61.94% | 48.35% |
| | CompassJudger-1-7B | 70.05% | 62.78% | 66.62% | 58.86% | 67.47% | 65.08% | 92.99% | 89.50% | 72.72% | 65.96% |
| | CompassJudger-1-14B | 58.94% | 44.62% | 55.09% | 40.76% | 59.66% | 52.90% | 90.87% | 86.61% | 63.22% | 51.37% |
| | CompassJudger-1-32B | 95.09% | 95.37% | 84.11% | 84.30% | 94.95% | 96.11% | 98.45% | 97.84% | 91.67% | 91.69% |
| | GPT-4o as Judge | 96.61% | 96.75% | 95.27% | 95.80% | 95.01% | 96.20% | 98.14% | 97.43% | 96.25% | 96.39% |
| | GPT-4o as Judge (CoT) | 97.10% | 97.23% | 95.41% | 95.88% | 95.63% | 96.63% | 99.56% | 99.38% | 96.85% | 96.95% |
| | xVerify-0.5B-I | 97.78% | 97.90% | 93.74% | 94.64% | **96.72%** | **97.49%** | 99.71% | 99.59% | 96.69% | 96.85% |
| | xVerify-1B-I | 97.22% | 97.35% | 94.76% | 95.45% | 96.06% | 96.97% | 99.71% | 99.59% | 96.77% | 96.91% |
| | xVerify-1.5B-I | 97.85% | 97.96% | 95.10% | 95.75% | 96.05% | 96.97% | 99.63% | 99.49% | 97.05% | 97.17% |
| | xVerify-2B-I | 97.93% | 98.02% | 95.06% | 95.71% | 96.06% | 96.97% | 99.78% | 99.69% | 97.09% | 97.21% |
| | xVerify-3B-Ia | 97.73% | 97.84% | 95.00% | 95.67% | 96.17% | 97.06% | 99.71% | 99.59% | 97.02% | 97.14% |
| | xVerify-3B-Ib | 97.31% | 97.41% | 95.65% | 96.18% | 96.38% | 97.23% | 99.78% | 99.69% | 97.17% | 97.27% |
| xVerify | xVerify-7B-I | 97.75% | 97.84% | 95.94% | 96.44% | 96.51% | 97.32% | 99.78% | 99.69% | **97.41%** | **97.50%** |
| | xVerify-8B-I | 97.92% | 98.02% | 95.34% | 95.97% | 96.05% | 96.97% | 99.71% | 99.59% | 97.17% | 97.29% |
| | xVerify-9B-C | **98.29%** | **98.38%** | 95.26% | 95.88% | 96.06% | 96.97% | 99.78% | 99.69% | 97.25% | 97.37% |
| | xVerify-9B-I | 97.43% | 97.53% | 95.75% | 96.27% | 96.06% | 96.97% | 99.78% | 99.69% | 97.19% | 97.29% |
| | xVerify-14B-Ia | 97.49% | 97.59% | 95.73% | 96.22% | 95.41% | 96.46% | 99.63% | 99.49% | 97.06% | 97.16% |
| | xVerify-14B-Ib | 97.67% | 97.78% | **96.10%** | **96.57%** | 95.74% | 96.72% | 99.71% | 99.59% | 97.31% | 97.40% |
| | xVerify-27B-I | 97.81% | 97.90% | 95.46% | 96.01% | 96.19% | 97.06% | 99.56% | 99.38% | 97.15% | 97.26% |
| | xVerify-32B-I | 97.81% | 97.90% | 95.88% | 96.31% | 96.18% | 97.06% | 99.71% | 99.59% | 97.32% | 97.40% |

## G.2 COMPUTATIONAL EFFICIENCY AND OPERATIONAL COST OF XVERIFY AND JUDGE MODELS

Table 25 displays the running time performance of the xVerify model and other judge models. Each model was evaluated using 200 randomly selected samples per question type from the generalization

Table 24: Evaluation Accuracy Results on the Generalization Set: All xVerify Models and Judge Models. The best performance in each column is shown in **bold**, and the second-best performance is underlined.

| Method Type | Method | Multiple Choice | | Math | | Short Answer | | Classification | | Overall | |
|---|---|---|---|---|---|---|---|---|---|---|---|
| | | F1 | Acc. | F1 | Acc. | F1 | Acc. | F1 | Acc. | F1 | Acc. |
| Judge Model | PandaLM-7B-v1 | 4.28% | 7.85% | 9.91% | 15.97% | 45.81% | 31.43% | 36.23% | 25.99% | 23.74% | 19.14% |
| | Auto-J-Bilingual-6B | 52.07% | 60.75% | 10.56% | 74.79% | 85.16% | 86.76% | 84.90% | 79.91% | 67.20% | 74.57% |
| | Auto-J-13B | 34.87% | 52.78% | 9.86% | 76.54% | 85.12% | 86.97% | 77.67% | 71.99% | 60.43% | 71.35% |
| | Prometheus-7B-v2.0 | 76.67% | 73.66% | 49.08% | 71.46% | 81.52% | 81.32% | 79.59% | 71.92% | 73.85% | 74.35% |
| | Prometheus-8x7B-v2.0 | 74.13% | 68.60% | 49.48% | 60.27% | 87.15% | 86.13% | 84.70% | 77.19% | 74.51% | 71.69% |
| | JudgeLM-7B-v1.0 | 60.22% | 45.71% | 12.71% | 15.40% | 72.15% | 62.51% | 86.11% | 76.18% | 59.11% | 46.38% |
| | JudgeLM-13B-v1.0 | 65.39% | 57.80% | 21.61% | 44.87% | 86.11% | 84.53% | 91.78% | 86.89% | 69.18% | 65.63% |
| | JudgeLM-33B-v1.0 | 46.99% | 45.10% | 20.31% | 39.99% | 71.34% | 66.69% | 41.92% | 33.36% | 46.06% | 46.01% |
| | CompassJudger-1-1.5B | 55.75% | 40.87% | 34.53% | 33.62% | 63.93% | 51.57% | 84.49% | 73.93% | 60.01% | 47.65% |
| | CompassJudger-1-7B | 74.31% | 65.20% | 38.27% | 39.89% | 88.99% | 88.15% | 93.29% | 89.29% | 73.47% | 67.47% |
| | CompassJudger-1-14B | 63.65% | 49.50% | 27.63% | 21.20% | 73.61% | 66.48% | 88.97% | 81.92% | 63.10% | 51.21% |
| | CompassJudger-1-32B | 92.93% | 92.32% | 72.05% | 84.91% | 96.81% | 96.86% | 98.05% | 97.05% | 91.90% | 92.04% |
| | GPT-4o as Judge | 95.86% | 95.38% | 87.91% | 94.76% | 97.46% | 97.49% | 98.67% | 97.98% | 96.03% | 96.18% |
| | GPT-4o as Judge (CoT) | 95.44% | 94.88% | 88.34% | 94.71% | 97.39% | 97.42% | 98.36% | 97.52% | 95.79% | 95.92% |
| xVerify | xVerify-0.5B-I | 96.49% | 96.10% | 80.00% | 91.94% | 96.95% | 97.00% | 99.03% | 98.53% | 95.29% | 95.53% |
| | xVerify-1B-I | 96.10% | 95.66% | 82.45% | 92.51% | 97.32% | 97.35% | 98.92% | 98.37% | 95.43% | 95.62% |
| | xVerify-1.5B-I | 96.76% | 96.38% | 83.58% | 93.12% | 97.46% | 97.49% | 98.88% | 98.29% | 95.85% | 96.03% |
| | xVerify-2B-I | 96.27% | 95.82% | 82.11% | 92.51% | **97.60%** | **97.63%** | 98.98% | 98.45% | 95.57% | 95.75% |
| | xVerify-3B-Ia | 96.44% | 95.99% | 86.10% | 94.25% | 97.31% | 97.35% | 99.03% | 98.53% | 96.11% | 96.27% |
| | xVerify-3B-Ib | 96.21% | 95.71% | 86.20% | 94.15% | **97.60%** | **97.63%** | 99.03% | 98.53% | 96.08% | 96.23% |
| | xVerify-7B-I | 96.16% | 95.66% | 87.86% | 94.87% | 97.45% | 97.49% | 98.93% | 98.37% | 96.22% | 96.37% |
| | xVerify-8B-I | 96.67% | 96.27% | 86.76% | 94.61% | 97.45% | 97.49% | 99.03% | 98.53% | 96.33% | 96.49% |
| | xVerify-9B-C | **97.00%** | **96.66%** | 87.08% | 94.71% | 97.45% | 97.49% | 98.98% | 98.45% | 96.45% | 96.61% |
| | xVerify-9B-I | 96.06% | 95.55% | 87.47% | 94.76% | 97.53% | 97.56% | **99.13%** | **98.68%** | 96.23% | 96.38% |
| | xVerify-14B-Ia | 96.11% | 95.60% | **90.20%** | **95.74%** | 97.32% | 97.35% | **99.13%** | **98.68%** | **96.53%** | **96.65%** |
| | xVerify-14B-Ib | 96.35% | 95.88% | 87.88% | 94.92% | 97.45% | 97.49% | 98.93% | 98.37% | 96.30% | 96.44% |
| | xVerify-27B-I | 96.01% | 95.49% | 85.64% | 93.99% | 97.32% | 97.35% | **99.13%** | **98.68%** | 95.93% | 96.09% |
| | xVerify-32B-I | 96.22% | 95.71% | 90.09% | 95.59% | 97.32% | 97.35% | 99.03% | 98.53% | 96.50% | 96.60% |

Table 25: Running Time Comparison of xVerify Models and Other Judge Models (200 Samples per Question Type). The best performance in each column is shown in **bold**, and the second-best performance is underlined.

| Method Type | Method | Multiple Choice (s) | Math (s) | Short Answer (s) | Classification (s) | Avg (s) |
|---|---|---|---|---|---|---|
| Judge Model | PandaLM-7B-v1 | 304.50 | 76.24 | 76.97 | 65.79 | 130.88 |
| | Auto-J-Bilingual-6B | 1,570.44 | 1,802.71 | 1,194.08 | 1,148.32 | 1,428.89 |
| | Auto-J-13B | 3,055.00 | 3,622.70 | 2,807.23 | 1,903.00 | 2,846.98 |
| | Prometheus-7B-v2.0 | 1,173.80 | 947.71 | 706.74 | 696.34 | 881.15 |
| | Prometheus-8x7B-v2.0 | 1,557.10 | 1,128.08 | 1,132.84 | 750.51 | 1,142.13 |
| | JudgeLM-7B-v1.0 | 551.88 | 469.10 | 394.57 | 348.05 | 440.90 |
| | JudgeLM-13B-v1.0 | 777.73 | 598.19 | 564.25 | 529.60 | 617.44 |
| | JudgeLM-33B-v1.0 | 1,041.83 | 1,018.37 | 789.80 | 762.99 | 903.25 |
| | CompassJudger-1-1.5B | 189.45 | 244.08 | 139.50 | 110.95 | 171.00 |
| | CompassJudger-1-7B | 163.96 | 568.72 | 450.20 | 80.58 | 315.87 |
| | CompassJudger-1-14B | 346.80 | 571.66 | 217.86 | 196.18 | 333.13 |
| | CompassJudger-1-32B | 147.53 | 258.10 | 133.59 | 152.11 | 172.83 |
| xVerify | xVerify-0.5B-I | 38.97 | 41.25 | 39.12 | 38.87 | 39.55 |
| | xVerify-1B-I | **33.91** | **36.63** | **33.44** | **33.47** | **34.36** |
| | xVerify-1.5B-I | 43.05 | 46.87 | 42.17 | 42.08 | 43.54 |
| | xVerify-2B-I | 38.44 | 73.16 | 39.29 | 37.38 | 47.07 |
| | xVerify-3B-Ia | 38.54 | 44.54 | 37.11 | 43.02 | 40.80 |
| | xVerify-3B-Ib | 46.93 | 53.58 | 106.06 | 47.84 | 63.60 |
| | xVerify-7B-I | 68.24 | 95.50 | 50.66 | 51.67 | 66.52 |
| | xVerify-8B-I | 78.06 | 61.57 | 45.34 | 46.82 | 57.95 |
| | xVerify-9B-C | 131.07 | 70.16 | 51.66 | 52.57 | 76.37 |
| | xVerify-9B-I | 54.20 | 69.91 | 49.41 | 51.06 | 56.15 |
| | xVerify-14B-Ia | 59.18 | 114.91 | 55.50 | 54.80 | 71.10 |
| | xVerify-14B-Ib | 61.17 | 145.19 | 116.43 | 57.55 | 95.09 |
| | xVerify-27B-I | 85.28 | 89.41 | 58.99 | 61.00 | 73.67 |
| | xVerify-32B-I | 131.05 | 98.99 | 64.74 | 67.45 | 90.56 |

set, with running times measured in seconds. This data provides insights into the computational efficiency of each model under uniform testing conditions, thereby facilitating a comparative analysis of their real-time processing capabilities and scalability in practical applications.

All models were executed on GPUs with identical configurations. Specifically, Prometheus-8x7B-v2.0, JudgeLM-33B-v1.0, CompassJudger-1-32B, xVerify-27B-I, and xVerify-32B-I were deployed on two GPUs for inference, while the remaining models were deployed on a single GPU. From Table 25, it is evident that all xVerify models exhibit an overall average runtime within 100 seconds,

whereas the overall average runtime for the other judge models exceeds 100 seconds. Moreover, for each question category, the models with the shortest evaluation times are the xVerify models. Thus, the xVerify models demonstrably surpass the other judge models in terms of evaluation efficiency.

Table 26 presents the evaluation costs incurred when employing GPT-4o as the judge, based on assessments of 200 randomly selected samples per question type, along with the overall expenditure. Apart from the prerequisite deployment overhead, the cost of invoking the xVerify models for evaluation is substantially lower than that of GPT-4o. Additionally, compared to GPT-4o, which relies on remote server deployment, the locally deployed xVerify models offer higher invocation efficiency. Taken together, these results underscore that the xVerify models outperform the other judge models in both usage cost and evaluation efficiency.

Table 26: Total costs (in USD) of GPT-4o as Judge (200 Samples per Question Type).

| Method | Multiple Choice ($) | Math ($) | Short Answer ($) | Classification ($) | Total ($) |
|---|---|---|---|---|---|
| **GPT-4o as Judge** | 0.31 | 0.66 | 0.24 | 0.27 | 1.48 |
| **GPT-4o as Judge (CoT)** | 0.55 | 1.00 | 0.42 | 0.48 | 2.45 |

### G.3 THE CONSISTENCY BETWEEN XVERIFY AND HUMAN EVALUATION

To evaluate the effectiveness of the xVerify, we conduct a comparative analysis of its consistency against human-annotated results. We evaluate the consistency between different models and human judgments across multiple evaluation datasets. The evaluation datasets and models utilized in this experiment are consistent with those employed in RL experiments (details in Appendix E). Figure 20 illustrates that xVerify consistently achieves a high degree of alignment with human evaluation across multiple datasets and in diverse scenarios.

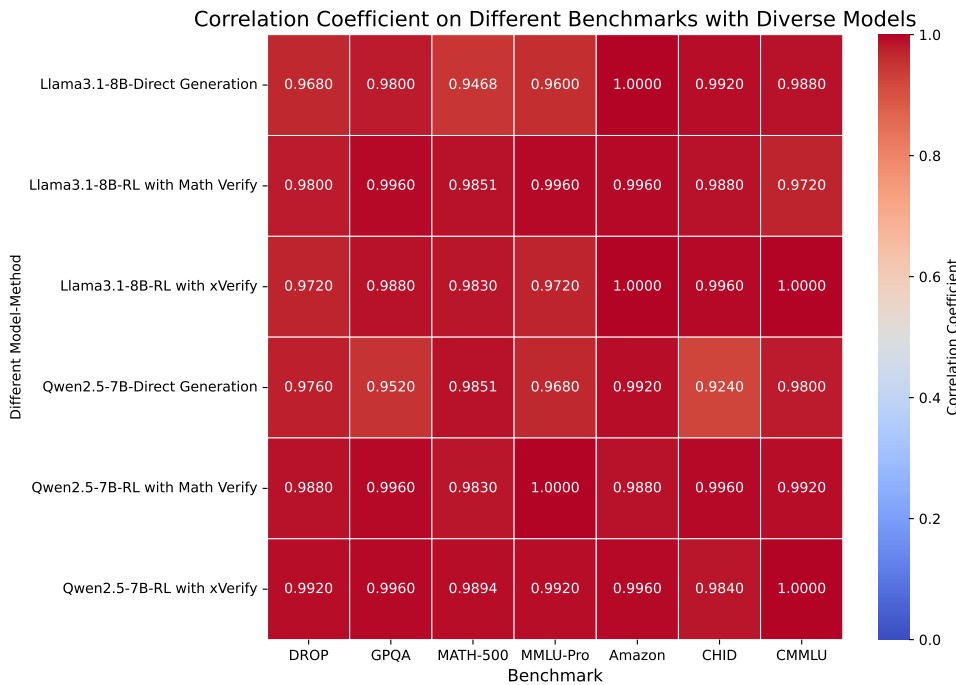

Figure 20: The Consistency between xVerify and Human Evaluation

### G.4 LEARNING CURVES OF QWEN2.5-7B AND LLAMA3.1-8B IN REINFORCEMENT LEARNING

Figures 21 and 22 present the learning curves of Qwen2.5-7B and Llama3.1-8B during reinforcement learning training. It is evident from these curves that using xVerify as the reward model results

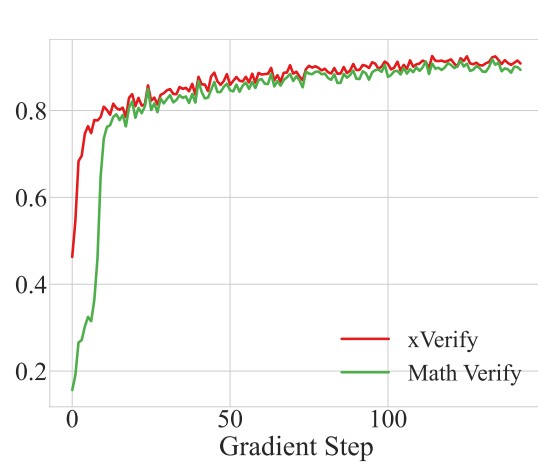

Figure 21: Learning Curves of Qwen2.5-7B in Reinforcement Learning

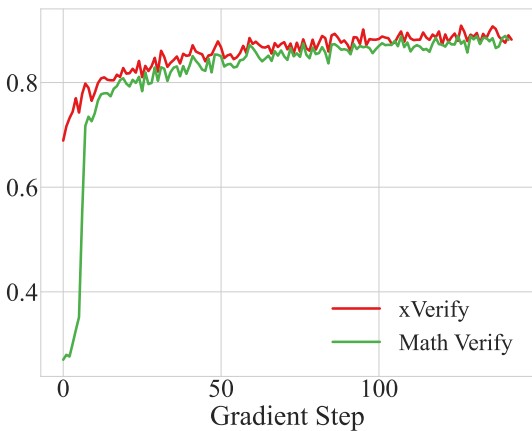

Figure 22: Learning Curves of Llama3.1-8B in Reinforcement Learning

in higher initial accuracy, indicating that xVerify provides more precise reward signals at the beginning of training. Moreover, the learning curves with xVerify converge faster, demonstrating that it delivers more accurate and reliable feedback, thereby significantly improving training efficiency. This is particularly important for larger-scale and broader reinforcement learning fine-tuning, where both training efficiency and reward signal quality are critical.

## G.5 FULL FINE-TUNING VS. QLoRA FOR XVERIFY MODELS

Table 27: Performance comparison of QLoRA and full fine-tuning on xVerify models.

| Model | Fine-tuning Type | Test Set | | Generalization Set | |
|---|---|---|---|---|---|
| | | F1 | Acc. | F1 | Acc. |
| Qwen2.5-0.5B-Instruct | QLoRA | 96.69% | 96.85% | 95.29% | 95.53% |
| | Full | 95.90% | 96.10% | 92.64% | 93.03% |
| LLaMA-3.2-3B-Instruct | QLoRA | 97.02% | 97.14% | 96.11% | 96.27% |
| | Full | 95.44% | 95.67% | 93.10% | 93.41% |

To quickly explore the impact of different fine-tuning strategies on xVerify models, we conducted a small-scale experiment comparing full fine-tuning with QLoRA on two representative models: Qwen2.5-0.5B-Instruct and LLaMA-3.2-3B-Instruct. Both models were fine-tuned on the VAR dataset, and their performance was evaluated on the test set as well as a held-out generalization set.

Table 27 summarizes the results. As shown, full fine-tuning achieves slightly lower overall performance compared to QLoRA. The decline is more pronounced on the generalization set, suggesting that full fine-tuning may be more prone to overfitting. In contrast, QLoRA maintains a better balance between test performance and generalization capability.

These findings indicate that, although full fine-tuning could in principle offer marginal gains under some conditions, in this small-scale exploration it does not outperform QLoRA. Additionally, QLoRA provides substantial computational savings, making it a more efficient choice for training instruction-tuned models. Overall, these results highlight that QLoRA is a practical and effective fine-tuning strategy for xVerify.

## G.6 COMPARISON OF XVERIFY MODELS WITH THEIR BASE MODELS

Table 28: Comparison of xVerify models with their base models on the test and generalization sets.

| Model | Test Set | | Generalization Set | |
|---|---|---|---|---|
| | F1 | Acc. | F1 | Acc. |
| Qwen2.5-0.5B-Instruct | 70.55% | 66.71% | 72.38% | 68.78% |
| xVerify-0.5B-I | **96.69%** | **96.85%** | **95.29%** | **95.53%** |
| LLaMA-3.2-1B-Instruct | 37.15% | 53.95% | 41.87% | 56.09% |
| xVerify-1B-I | **96.77%** | **96.91%** | **95.43%** | **95.62%** |
| LLaMA-3.2-3B-Instruct | 85.22% | 84.99% | 83.79% | 83.32% |
| xVerify-3B-Ia | **97.02%** | **97.14%** | **96.11%** | **96.27%** |
| Gemma-2-9B-it | 83.31% | 81.89% | 82.20% | 80.61% |
| xVerify-9B-I | **97.19%** | **97.29%** | **96.23%** | **96.38%** |

To assess the effectiveness of VAR-based fine-tuning, we conducted a systematic evaluation of four xVerify variants and their corresponding base models on both the test and generalization sets, using identical evaluation settings. As shown in Table 28, the fine-tuned xVerify models achieve over 10 percentage points of gain in both F1 and accuracy compared to their base models. Notably, xVerify-0.5B-I and xVerify-1B-I each deliver improvements of 20%–50% on these metrics. These results

demonstrate that VAR-based fine-tuning substantially enhances the evaluation capabilities of the base models.

## G.7    Evaluation without the original question

Table 29: Performance of xVerify models with and without access to the original question.

| Dataset | Model | Setting | F1 | Acc. |
|---------|-------|---------|-----|------|
| Test | xVerify-0.5B-I | with question | 96.69% | 96.85% |
| | | w/o question | 96.53% | 96.73% |
| | xVerify-3B-Ia | with question | 97.02% | 97.14% |
| | | w/o question | 96.09% | 96.30% |
| Generalization | xVerify-0.5B-I | with question | 95.29% | 95.53% |
| | | w/o question | 95.05% | 95.33% |
| | xVerify-3B-Ia | with question | 96.11% | 96.27% |
| | | w/o question | 95.55% | 95.79% |

In real-world evaluation scenarios, it is often the case that only the LLM's response and the reference answer are available, without access to the original question. This setting arises, for example, when xVerify is used as a reward model in reinforcement learning.

To assess whether xVerify can maintain strong accuracy and generalization under these conditions, we evaluated **xVerify-0.5B-I** and **xVerify-3B-Ia** on both the test set and the generalization set, with and without providing the original questions. In the modified setup, we removed only the parts of the prompt referencing the question, while keeping all other configurations identical.

As shown in Table 29, both models exhibit virtually no performance drop across datasets, demonstrating that xVerify retains robust decision-making capability and generalization even in the absence of the original question. This suggests that xVerify's core judgment mechanism relies primarily on the equivalence between the model's output and the reference answer, while the original question serves only as auxiliary context rather than essential input.

## H    Limitations

In this work, we focus on building a more accurate and efficient answer equivalence verifier, particularly for assessing the equivalence between the outputs of reasoning models and reference answers. However, the current xVerify model is not yet capable of fully replacing all evaluation methods and reward models. On the one hand, xVerify is currently tailored to objective questions with definitive answers and lacks effective optimization for subjective questions. On the other hand, xVerify cannot yet serve as a complete substitute for general-purpose reward models, as such models typically require only the input question and the output from the policy model to generate reward signals, whereas xVerify still depends on the availability of reference answers. Addressing these limitations may require future efforts to enhance xVerify's capability in scenarios without reference answers, such as those involving subjective questions. This could involve the development of new task settings and corresponding datasets for training and optimizing the xVerify model.

