# OpenReview forum: "xVerify: Efficient Answer Verifier for Reasoning Model Evaluations"
_ICLR.cc/2026/Conference — ICLR 2026 Conference Withdrawn Submission_

### Official Review · Reviewer_eRsX · 2025-10-30

**Soundness:** 3
**Presentation:** 2
**Contribution:** 2
**Rating:** 4
**Confidence:** 4

**Summary:**

This paper aims to construct a unified verifier for various subjective Q&As. The authors start by creating the Verify Answer for Reasoning (VAR) dataset, followed by training a series of xVerify models using the VAR datasets based on LLMs of different series of sizes.  The authors first conduct proof-of-concept experiments with RL to demonstrate the effectiveness of the proposed models.

**Strengths:**

- The paper is well-written and easy to understand.
- The idea is easy and straightforward.
- I like the experiments of RL, which should be the key to training a unified verifier.

**Weaknesses:**

- About problem definition:
  - In lines 150-154, the authors claim to "extract answers", which, however, is actually conducting Best-of-N.
  - I wonder why the authors spend half a page discussing different domain-specific verifiers in Eqn. 1-4, which is neither practical (i.e., 1), you cannot know the question type ahead of time; 2) you cannot guarantee subjective questions belong to one of the 4 question types. nor necessary, since it has no connection with the dataset annotation (mainly relying on GPT-4o and humans) and xVerify models.
- About dataset construction:
  - In Sec. 4.1.3, why do you decide to do two-rounds of GPT-4o prompting?
  - If the two rounds are connected, I cannot see it from the prompt templates.
  - If the two rounds are independent and only for confidence filtering, why do you choose to use different prompts?
- About VAR datasets:
  - Just to make sure, the training targets in VAR are simply Correct/Wrong, right? Do you collect any reasoning texts?
  - If so, when you train xVerify, you simply fine-tune an LLM to conduct a binary classification problem using SFT, right?
  - Can you explain more about how your generalization sets are different from the training/testing sets? At least I cannot see it in Tables 1-2.
- About xVerify:
  - When you train on VAR and then evaluate on VAR, the biggest concern is data leakage.
  - Although the authors claim to do data cleaning and separation (e.g., the generalization set), the results in Tables 1-2 seem to demonstrate different conclusions.
  - All xVerify models' results are over 95% for both F1 and accuracy for both the testing and generalization sets, spanning from the 0.5B to the 32B variants.
  - Model sizes do not matter. Base models do not matter. And a 0.5B variant performs better than GPT-4o LLM-as-a-judge. The empirical results fully demonstrate the strong correlation between the training and testing sets.
- Overall, a unified verifier is important, but the key is generalizability.

**Questions:**

Check the Weakness part for details.

---

### Official Review · Reviewer_sQTd · 2025-10-31

**Soundness:** 3
**Presentation:** 2
**Contribution:** 2
**Rating:** 2
**Confidence:** 4

**Summary:**

The paper proposes xVerify, a lightweight “judge” model family trained to determine answer equivalence for objective reasoning tasks, addressing two pain points in evaluating reasoning models: (i) extracting the final answer from long chains of thought and (ii) judging semantic/mathematical equivalence to a reference answer. The authors also contribute VAR, a supervised dataset aggregating responses from 19 LLMs across 24 benchmarks, labeled via multiple rounds of GPT-4o plus human verification. They fine-tune models on their proposed dataset and achieve high judge accuracy on generalization sets; notably the 0.5B model beats most baselines (including some 32B judges), and larger versions can even surpass GPT-4o on their evaluation. They also show that xVerify as a reward model in RL can improve average downstream task accuracy. Their method is aimed at providing one model for unified equivalence assessment combining mathematical, natural-language, and symbolic normalization/matching.

**Strengths:**

The problem of evaluation is important and the paper does a good job in using diverse model family, sizes and various datasets to comprehensively evaluate their approach. A lot of past work has been done on using LLM-as-a-judge (which authors already cite) but has mostly been focused on subjective tasks as objective tasks are generally taken care via rule-based approaches. Several baselines are considered for comparison showing how xVerify models are better than many other approaches. It is also laudable that they do RL training using their proposed judge since the ultimate application of measurement (evaluation) is to provide signal for learning which the authors demonstrate. The results are strong with x-Verify models having >= 95% F1/acc scores.


The paper does not tackle a totally new problem but takes a data-focused approach to improve LLM-judges. The VAR dataset could also be a very useful resource for the open-source community (assuming it will be released publicly if accepted). The paper is dense and comprehensive with all the details in the Appendix but is mostly easy to follow.

**Weaknesses:**

Main Concerns:

1. I don't fully agree with the premise of the paper that reasoning models pose extra challenges for evaluation as all the extra thinking steps is often enclosed inside <think> tags which can be stripped away from final evaluation. For example, I find the claims in L45-L47 unjustified. The claims are not backed by any proper evidence -- the only paper cited is Chang et al. came out in 2023 well before the release of any reasoning models (o1 came out in Sept'24).

2. Inter-Annotator Agreement: Almost all the technical details are deferred to appendix like dataset composition, how annotation was done, what exactly was asked to the annotator (judge/human). Neither inter-annotator agreement rate between humans nor GPT4o-annotator agreement with humans is reported thus questioning the credibility of GPT-4o.

3. Response Generation: It is well-known that models are asked to "Provide final answer in \\boxed{} format" for math/MCQ evaluations and also trained via such prompts. Thus, this should be used in when generating responses (when constructing the dataset) otherwise it would lead to unfair comparison between evaluation frameworks downstream. This way of prompting is well-established and works well in practice as also used in frontier evals like Humanity Last Exam [4].

    It is not possible to assess the actual benefit of xVerify models when the response being evaluated is not properly sought at the first place (i.e., the test/generalization split of VAR dataset since it does not reflect "exactly" how LLMs are prompted in LM-eval-harness or LightEval frameworks). I would recommend the authors to re-run response generation in well established template as followed by LightEval (with CoT prompting).

    Classification can be viewed as special case of MCQ essentially. Thus, all formats (MCQ, math, classification) except short answer can be handled via \\boxed format which is well established in the community. For short text answer, the prompt can be to "Provide final answer in \<answer\> \</answer\> tags." This has been done in recent work like training DeepSeek-R1 [1].

4. In Section 5.2, what was the training dataset for RL? How is the evaluation actually performed? It does not make sense to "compare" with direct generation as direction generation as per my understanding is without any training. The fair comparison is only with math-verify and the improvement there is less than 1% so there hardly seems to be any benefit of using xVerify over math-verify. Further, xVerify would require loading a judge model in memory compared to math-verify which would also require more GPU memory/time (although loading a 0.5B model should not be an issue).

5. Continuing on RL training, in the referenced Fig. 22, why is the reward between xVerify and math-verify so different at step 0 (which is before training began)? All methods should start from the same starting point assuming the underly evaluation is consistent across all.

6. If math-verify can almost match x-verify (based on the results of RL-training), then the large gain in accuracy reported in Table 1 seems confusing? I would appreciate if the authors can clarify how exactly is the evaluation being performed for the final results table/figure.

7. More baseline: While many LLM-as-a-judges have been compared, all of them are quite old (For example, while JudgeLLM got published in ICLR'25, it came out on Arxiv in 2023 and is based on models released > 1.5 year ago). I would also like to see whether other recent models (both reasoning/non-reasoning eg: Qwen3-4B) can act off-the-shelf as good judges as shown in a recent study [3] which also focuses on using judges for objective answer assessment.

8. Important Related Work: No comparison is made with existing general-verifier model used in recent work [2] where the authors also fine-tune small 1.5B judge for their tasks (also leading to high accuracy downstream with RL) and judge model is shown to have high agreement with Gemini 2 Flash. [2] is highly similar with this paper and deserved to be compared.


Typos:

L-070: "We fine-tune xVerify on a variety of base models" -> "We finetune a variety of base models on xVerify"

In Fig.1, LLM-response second box corresponding answer should be (B) instead of (A)?

L311-L312: Spacing is weird. Please increase spacing.


References:

[1] Guo, D., Yang, D., Zhang, H., Song, J., Zhang, R., Xu, R., ... & He, Y. (2025). Deepseek-r1: Incentivizing reasoning capability in llms via reinforcement learning. arXiv preprint arXiv:2501.12948.

[2] Ma, X., Liu, Q., Jiang, D., Zhang, G., Ma, Z., & Chen, W. (2025). General-reasoner: Advancing llm reasoning across all domains. arXiv preprint arXiv:2505.14652.

[3] Chandak, N., Goel, S., Prabhu, A., Hardt, M., & Geiping, J. (2025). Answer Matching Outperforms Multiple Choice for Language Model Evaluation. arXiv preprint arXiv:2507.02856.

[4] Phan, L., Gatti, A., Han, Z., Li, N., Hu, J., Zhang, H., ... & Wykowski, J. (2025). Humanity's last exam. arXiv preprint arXiv:2501.14249.

**Questions:**

Q1: How is the accuracy in Table 1 calculated? Is it calculated by comparing the judgement to human judgement?

Q2: In L152, a scoring function is defined but nothing is mentioned about how the suitability is calculated or assessed for each candidate. No details is provided in this selection. I would appreciate if authors can elaborate here.

Q3: For annotation, was ethics board IRB approval taken to hire annotators? While I have not flagged the paper for ethical concerns, nothing was mentioned regarding IRB approval in the paper.

Q4: What do the suffix "-I", "-Ia", "-Ib" for xVerify models in Table 1 and 2 mean?

Q5: For different judge models, was different ways of prompting tried? Like providing in-context examples?

---

### Official Review · Reviewer_cERW · 2025-10-31

**Soundness:** 3
**Presentation:** 3
**Contribution:** 2
**Rating:** 2
**Confidence:** 4

**Summary:**

> This paper may have violated the Double-blind submission and Anonymity policy of ICLR. The huggingface link of the authors' lab has been
listed in their [repo](https://anonymous.4open.science/r/xVerify-5702):

```html
<a href="https://huggingface.co/IAAR-Shanghai">
    <img alt="Huggingface" src="https://img.shields.io/badge/🤗 Huggingface-Models-orange.svg">
</a>
```


The paper proposes xVerify, a lightweight and efficient verifier for reasoning-based tasks.
It introduces the VAR dataset covering 19 LLMs and 24 benchmarks, and trains models of different sizes to predict whether an answer is correct.
Evaluation demonstrates xVerify's two key capabilities: 1) It exhibits superior accuracy and robustness against existing methods on both in-domain and out-of-distribution benchmarks; 2) It shows advantages when used as a reward model for RL training.

**Strengths:**

1. The proposed xVerify framework provides an efficient and scalable solution, achieving strong performance across diverse reasoning benchmarks.
2. The introduction of the VAR dataset contributes valuable large-scale resources for objective answer verification.
3. The experimental evaluation is extensive, covering multiple model sizes, in-domain and out-of-domain tests, and reinforcement learning scenarios.

**Weaknesses:**

1. The novelty of the work is limited. The main contribution lies in building the VAR dataset and fine-tuning existing LLMs for verification. The paper does not introduce new architectures or learning mechanisms, and similar evaluator training approaches have been explored in prior works such as xFinder[1] and CompassJudger[2]. A clearer explanation of what makes xVerify distinct or innovative would strengthen the paper (Q1-Q2).

2. The implementation details of the core verification components are insufficiently described, making it unclear how symbolic and semantic alignments are realized (Q3-Q4).

3. While the results are strong, the paper does not explain *why* xVerify outperforms rule-based or LLM-based judge models (Q5).

**Questions:**

1. Since the VAR dataset is labeled only with *Correct* or *Incorrect* judgments, how does this binary annotation handle borderline or partially correct cases, especially in reasoning-intensive answers? Would introducing graded or confidence-based labels better reflect nuanced correctness and reduce evaluation bias?

2. The paper does not include a clear analysis of the VAR dataset’s quality or consistency. Could the authors report inter-annotator agreement metrics or other measures to validate annotation reliability?

3. In Section 3 (*Problem Definition*), Equations (1)–(3) introduce the sub-functions $\psi_{\text{math}}$, $\psi_{\text{nl}}$, and $\psi_{\text{sym}}$, but their concrete implementations are not clearly described. Are these functions realized through symbolic or string-based normalization methods (e.g., regular expressions, symbolic computation), or are they implemented via LLM-based semantic alignment?

4. In Section 3 (*Problem Definition*), Equations (2), for natural language equivalence, could the author clarify how $\varphi_{\text{nl}}^{\text{align}}$ measures semantic equivalence (e.g., embedding similarity or LLM scoring) and how the threshold $\tau$ is determined? Since $\tau$ critically affects the correctness boundary, it would be valuable to include sensitivity analyses or ablation studies regarding its influence on xVerify’s performance.

5. While xVerify shows strong results, the paper does not clearly explain *why* it outperforms rule-based methods (e.g., RegEx) or other LLM-based judges. Could the authors provide more analysis of the main factors driving this improvement? In addition, how does xVerify differ in core methodology and design from recent evaluator models such as xFinder [1] and CompassJudger [2]?

   [1] Qingchen Yu et al., *xFinder: Large Language Models as Automated Evaluators for Reliable Evaluation*, ICLR 2025.

   [2] Taolin Zhang et al., *CompassJudger-2: Towards Generalist Judge Model via Verifiable Rewards*, arXiv preprint arXiv:2507.09104, 2025.

---

### Official Review · Reviewer_i4jB · 2025-10-31

**Soundness:** 2
**Presentation:** 2
**Contribution:** 2
**Rating:** 2
**Confidence:** 3

**Summary:**

This paper proposed xVerify, an efficient answer verifier for objective reasoning tasks that extracts a final answer from long LLM responses and checks equivalence via math, symbol, and natural-language comparators; it also introduces the VAR dataset built from 19 LLMs across 24 benchmarks to train and evaluate such judges.

**Strengths:**

1. high performance and efficiency, showing 95% f1 with 0.5B parameters
2. the proposed dataset can benefit other research in the community

**Weaknesses:**

1. I am curious how the process-involved reward can be superior than the pure rule-based reward. And why the process involved reward is important in the multiple choice questions, which are very easy to verify via answer correctness.
2. The final output of xverify only focous on the final answer, not the soudness of the reasoning chain. The output is a binary label, also doesn't provide more fine-grained information regarding the correcctness of the reasoning chain.
3. The scope is limited, xVerify targets objective tasks and needs reference answers; it does not handle subjective evaluation and is not a drop-in reward model when ground truth is absent.
4. xVerify decides equivalence by first normalizing answers, and compare the semantic alignment score. If the normalizer cannot cover some edge case, the model can be hacked.

**Questions:**

please see weaknesses above

---

### Note · Authors · 2025-11-12

I have read and agree with the venue's withdrawal policy on behalf of myself and my co-authors.